# Arid3a regulates nephric tubule regeneration via evolutionarily conserved regeneration signal-response enhancers

Nanoka Suzuki[1], Kodai Hirano[1], Hajime Ogino[2], Haruki Ochi[1]*

[1]Institute for Promotion of Medical Science Research, Yamagata University, Faculty of Medicine, Yamagata, Japan; [2]Amphibian Research Center, Hiroshima University, Higashi-hiroshima, Japan

**Abstract** Amphibians and fish have the ability to regenerate numerous tissues, whereas mammals have a limited regenerative capacity. Despite numerous developmental genes becoming reactivated during regeneration, an extensive analysis is yet to be performed on whether highly regenerative animals utilize unique *cis*-regulatory elements for the reactivation of genes during regeneration and how such *cis*-regulatory elements become activated. Here, we screened regeneration signal-response enhancers at the *lhx1* locus using *Xenopus* and found that the noncoding elements conserved from fish to human function as enhancers in the regenerating nephric tubules. A DNA-binding motif of Arid3a, a component of H3K9me3 demethylases, was commonly found in RSREs. Arid3a binds to RSREs and reduces the H3K9me3 levels. It promotes cell cycle progression and causes the outgrowth of nephric tubules, whereas the conditional knockdown of *arid3a* using photo-morpholino inhibits regeneration. These results suggest that Arid3a contributes to the regeneration of nephric tubules by decreasing H3K9me3 on RSREs.
DOI: https://doi.org/10.7554/eLife.43186.001

*For correspondence:
harukiochi@med.id.yamagata-u.ac.jp

Competing interests: The authors declare that no competing interests exist.

## Introduction

Kidneys are necessary for the maintenance of homeostasis in vertebrates. The functional loss of this organ leads to severe defects in systems vital to all vertebrates, including human. During evolution, vertebrates evolved the following three complex kidney structures: pronephros, mesonephros, and metanephros. Metanephros is the adult kidney in higher vertebrates, such as humans and mice, whereas mesonephros is the adult kidney in fish and amphibians. Pronephros is the simplest and earliest kidney form (*Jones, 2005*; *Desgrange and Cereghini, 2015*). Despite the different levels of complexity of these three kidney types, their functional unit, the nephron, does not differ much (*Brändli, 1999*; *Lienkamp, 2016*). In human, there are approximately one million nephrons per kidney (*Saxén and Sariola, 1987*). Each nephron is composed of a glomerulus, a filtering component, and a nephric tubule, which is divided into the following four basic domains: proximal tubule, loop of Henle, distal tubule, and connecting tubule (*Saxén and Sariola, 1987*). According to molecular studies, the nephrons of the pronephros in *Xenopus laevis* can be subdivided into four distinct segments that are homologous to the segments of the metanephric nephrons of mammals (*Raciti et al., 2008*). Owing to these similarities, *X. laevis* is considered a suitable model for studying renal regeneration for application to human.

Gentamicin-induced nephropathy and partial nephrectomy showed that numerous vertebrates have the ability to repair damage to nephric structures, although this regenerative ability diverges between species (*Nonclercq et al., 1992*; *Diep et al., 2011*; *Zhou et al., 2010*; *Caine and Mclaughlin, 2013*). In mammals, nephrons contain mature tubular epithelial cells that have the capacity to regenerate following acute kidney injury (*Maeshima et al., 2014*). These epithelial cells rapidly lose

their brush border, dedifferentiate into mesenchymal-like cells, and then migrate into regions where cells are damaged (*Maeshima et al., 2014*). Recent cell lineage studies have also suggested that renal stem cells and progenitor cells were found in the metanephric mesenchyme, limb of Henle's loop, and distal convoluted tubule in mice (*Kobayashi et al., 2008*; *Barker et al., 2012*). Although mammalian nephrons possess cells that contribute to repair after injury, the regenerative capacity is restricted to the reconstruction of nephric epithelial cells in damaged regions (*Maeshima et al., 2015*). In contrast to mammals, which merely reconstitute tubular epithelial cells after injury, previous studies showed that *X. laevis* and zebrafish regenerate fully coiled and functional nephric tubule architecture after severe damage (*Diep et al., 2011*; *Caine and Mclaughlin, 2013*). In zebrafish, the transplantation of *lhx1a*-positive or *six2* mesenchymal cells, which are considered stem cells, into adult fish in which the kidney has been injured by gentamicin reconstructs functional nephrons (*Diep et al., 2011*). *X. laevis* also regenerates the functional pronephros that can uptake albumin again after the mechanical loss of proximal tubules (*Caine and Mclaughlin, 2013*). Both amphibians and fish have a high regenerative capacity, but their regenerative mechanisms appear to differ. Zebrafish use kidney stem cells to repair the functional nephrons, whereas *X. laevis* appear to use remaining tubule cells to regenerate the functional pronephros (*Diep et al., 2011*; *Caine and Mclaughlin, 2013*). Thus, despite the different regenerative mechanisms, numerous vertebrates have the ability to repair injured nephrons.

Recently, studies have established protocols for generating the nephric structure derived from induced pluripotent cells (iPSCs) and embryonic stem (ES) cells by the combined application of the Wnt activator and other signaling factors such as BMP4 (*Xia et al., 2013*; *Takasato et al., 2014*; *Takasato et al., 2015*). In addition, direct reprogramming from fibroblasts into renal tubular epithelial cells, called induced renal tubular epithelial cells (iRECs), has been successfully achieved using a combination of transcription factors whose expression in the kidney is evolutionarily conserved between mice and *Xenopus* (*Kaminski et al., 2016*; *Lienkamp, 2016*). However, although the molecular factors for the construction of nephrons *in vivo* and *in vitro* have been identified, the molecular mechanisms that allow the reactivation of developmental genes during regeneration in highly regenerative animals remain unclear.

Genetic studies have shown that BMP and canonical Wnt and Fgf signals first specify intermediate mesoderm to form the pronephric primordium, followed by epithelialization, during kidney development (*Krause et al., 2015*). In addition to signaling factors, transcription factors such as Osr1, Osr2, Pax8, Hnf1b, Lhx1, and WT1 also coordinately regulate kidney formation (*Bouchard, 2004*; *Lienkamp, 2016*). These developmental genes for the kidney are evolutionarily conserved between pronephros formation in *Xenopus* and the more complex nephrons in mammalian meso- and metanephros (*Lienkamp, 2016*). During the regeneration of nephrons in zebrafish, transplanted *lhx1a*-positive mesenchymal cells begin to express developmental genes such as *pax2a*, *wt1a*, and *fgf8a* (*Diep et al., 2011*). In order to use developmental genes during regeneration, the genes have to be expressed again in regenerating tissues, which requires *cis*-regulatory elements called enhancers. Enhancers are known to determine the tissue specificity and timing of gene expression, and over 43,000 active enhancers have been identified in the human genome alone (*Andersson et al., 2014*). To date, numerous enhancers for tissue and organ development have been identified by the deletion mapping of upstream genomic regions and/or candidate screening using the epigenetic landscape (*Kleftogiannis et al., 2016*), whereas only a few enhancers for tissue regeneration have been reported, such as tissue regeneration enhancer elements for zebrafish heart reconstruction (*Kang et al., 2016*). Although it is generally assumed that the genome of vertebrates contains more enhancers that promote gene expression in regenerating tissues, the difficulty in identifying the *in vivo* function of enhancers prevents us from revealing the *cis*-regulatory architecture for regeneration. We have previously established an *in vivo* enhancer mapping system using *X. laevis* transgenesis (*Ogino et al., 2008*; *Ochi et al., 2012*; *Suzuki et al., 2015*). Here, we extended this strategy for identifying enhancers for the regeneration of nephric tubules and for examining the transcriptional mechanisms that regulate the enhancer activities during regeneration. We first found that *lhx1* expression was induced within 24 hr after the surgical removal of nephric tubules. In order to explore the induction mechanisms of *lhx1* in regenerating nephric tubules, we screened enhancers in genomic loci using a transgenic mapping system and found that genomic elements that are conserved from fish to human, called regeneration signal-response enhancers (RSREs), were activated in regenerating nephric tubules. A putative binding motif of Arid3a, a member of the AT-rich

interaction domain family, is commonly found in RSREs for *lhx1*. We found that Arid3a directly binds to RSREs and Arid3a with H3K9me3 demethylases Kdm4/Jmjd2 reducing H3K9me3 levels on RSREs. We further found that the conditional knockdown of *arid3a* during regeneration suppressed the regeneration of nephric tubules. We now demonstrate that Arid3a is required for the regeneration of nephric tubules through the RSREs.

## Results

### Proximal tubules and intermediate tubules have different regenerative capacities

The pronephric kidneys of *X. laevis* become functional from Nieuwkoop and Faber stages 37–38. A previous study showed that proximal tubules that have been partially nephrectomized at stages 37–38 can regenerate functional nephrons (*Raciti et al., 2008*; *Caine and Mclaughlin, 2013*). Since we have previously established the *Xla.Tg(Xtr.pax8:EGFP)* transgenic line that can be used to trace nephric tubule formation, we first confirmed the regenerative process of nephric tubules using live imaging of stage 37 transgenic embryos (*Ochi et al., 2012*). Our careful nephric tubule surgery did not damage the glomus, which is situated medially to the tubules displaying *podocin* expression (*Figure 1—figure supplement 1*). When the proximal tubules were partially nephrectomized, a coiled structure appeared at around 24 hr (*Figures 1A, B and E*). This coiled tubule continued to extend from 24 to 48 hr, and then the regenerated tubules reconstructed the nephron between 72 and 120 hr (*Figure 1B and E*). In contrast, completely nephrectomized proximal tubules failed to form a coiled structure, whereas the remaining tubules, the intermediate tubules, continued to extend from the back to the front (*Figure 1C and E*). Conversely, when the intermediate tubules were completely removed, no tubule extension was observed (*Figure 1D and E*). Thus, although *Xla.Tg(Xtr.pax8:EGFP)* cannot capture all of the cells that contribute to the reconstruction of nephric tubules, our results indicate that the proximal tubules are essential for the regeneration of the coiled structure of the nephric duct. In addition, the intermediate tubules also have the capacity to regenerate nephric tubules, although this capacity is limited compared with that of the proximal tubules. It has been shown in a previous study that apoptotic cells and the expression of matrix metalloproteinase nine dramatically increase within 12 hr after nephrectomy, and these events continue until 24–48 hr (*Caine and Mclaughlin, 2013*). Our observation that regenerative tubules exhibit a coiled structure at around 24–48 hr together with previous findings suggests that the genetic mechanisms that occur within 48 hr after nephrectomy are crucial for nephron regeneration.

### *lhx1* expression appears in regenerating nephrons immediately after nephrectomy

In order to explore the molecular mechanisms behind the regeneration of nephric tubules, we examined the expression of the transcription factors *osr1*, *osr2*, *lhx1*, *six2*, *hnf1b*, *hnf4a*, *pax2*, *pax8*, and *wt1* in regenerating nephric tubules. Among them, *osr1*, *osr2*, and *lhx1*, together with *pax8*, are known to induce ectopic nephrons when overexpressed in *Xenopus* embryos (*Seufert et al., 1999*; *Carroll and Vize, 1999*; *Tételin and Jones, 2010*), and *lhx1*- or *six2*-expressing mesenchymal cells can reconstruct functional nephrons in kidney-injured zebrafish (*Diep et al., 2011*). The *in situ* hybridization analysis showed that the expression of *lhx1*, *pax8*, *hnf4a*, *hnf1b*, and *osr2* appeared within 24 hr after the nephrectomy (*Figures 2A, D, G and J*, *Figure 2—figure supplement 1*). Such expression became stronger around 48 hr (*Figures 2B, E, H and K*, *Figure 2—figure supplement 1*). Then, the expression of *lhx1* disappeared at 96 hr, while the expression of *pax8* was still detected in the nephric tubules (*Figures 2C, F, I and L*). In contrast, the expression of *pax2* and *osr1* appeared around 48 hr, and that of *six2* was not detected in regenerating nephric tubules (*Figure 2M–2R*, *Figure 2—figure supplement 1*, data not shown). Although it is possible that primary induced *lhx1*-, *pax8*-, *hnf4a*-, *hnf1b*-, and *osr2*-expressing cells are not completely consistent with cells observed in *Xla.Tg(Xtr.pax8:EGFP)* regenerating nephric tubules, the immediate induction of *lhx1*, *pax8*, *hnf4a*, *hnf1b*, and *osr2* after nephrectomy suggests that the loci of these genes contain enhancers of injury response.

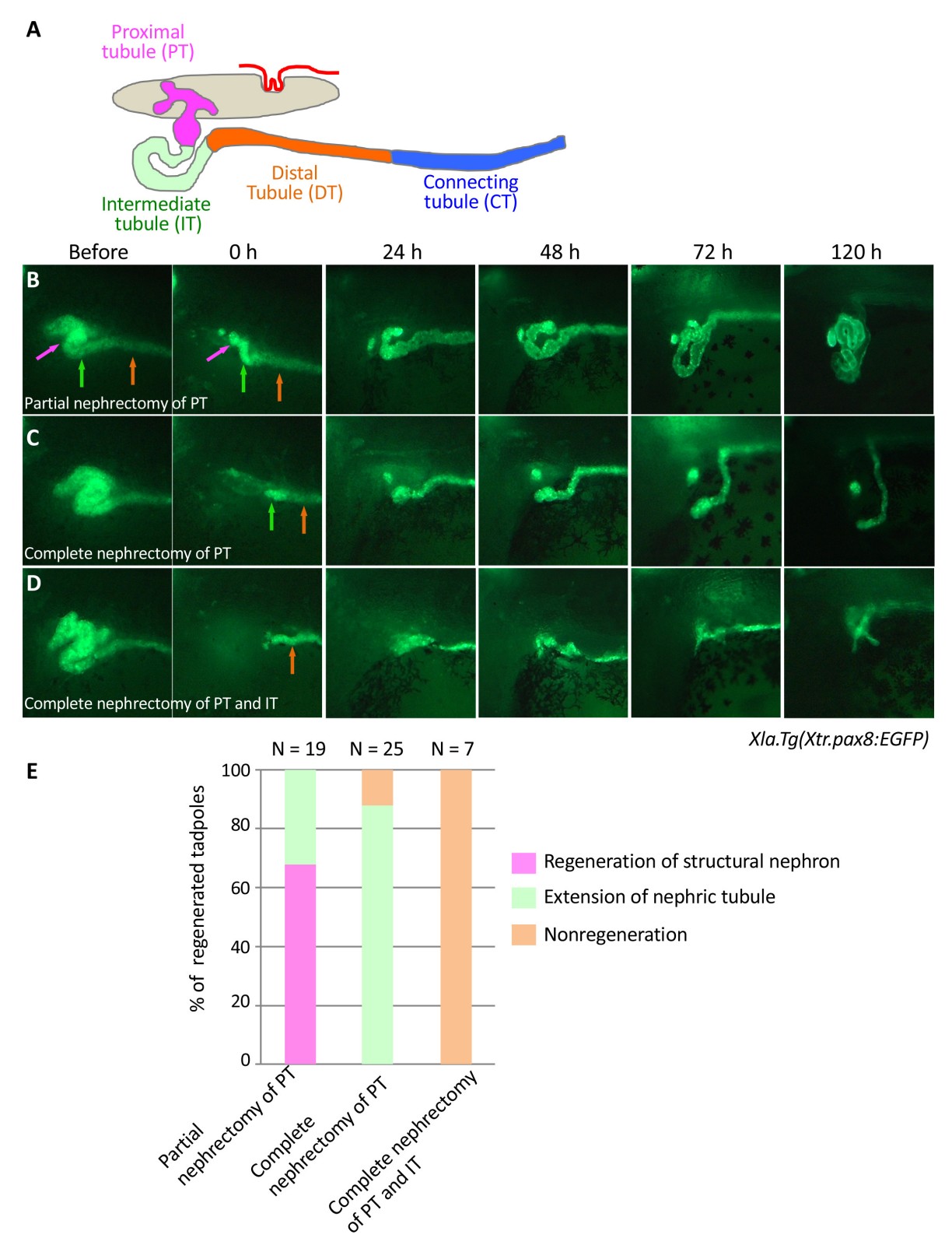

*Xla.Tg(Xtr.pax8:EGFP)*

**Figure 1.** Live imaging of nephric tubules using transgenic *X. laevis*. (**A**) A schematic image of *X. laevis* nephron. PT: proximal tubule; IT: intermediate tubule; DT: distal tubule; CT: connecting tubule. Magenta arrows: PT; green arrows: IT; orange arrows: DT. (**B**) Representative regeneration pattern of partially resected proximal tubules. Proximal tubules regenerate a coiled structure. (**C**) Regeneration pattern of completely resected proximal tubules. The remaining intermediate tubules extend, but no coiled structure is regenerated. (**D**) Regeneration pattern of completely resected intermediate

*Figure 1 continued on next page*

*Figure 1 continued*

tubules. No extension of tubules is observed. (E) Statistics of the regeneration pattern. The statistics of three independent experiments are summarized.

DOI: https://doi.org/10.7554/eLife.43186.002

The following figure supplement is available for figure 1:

**Figure supplement 1.** Podocytes are not injured by the surgical removal of nephron tubules.

DOI: https://doi.org/10.7554/eLife.43186.003

## Mapping of RSREs

The expression of *lhx1*, *pax8*, and *hnf1b* was immediately induced after injury. It has been shown in a previous study involving simple mRNA injection using *Xenopus* embryos that Lhx1 with Pax8 can induce ectopic pronephron structures (*Carroll and Vize, 1999*). In addition, tissue-specific *lhx1* knockout using CRISPR/Cas9 genome engineering clearly showed that Lhx1 is required for the development of nephric tubules in *Xenopus* (*DeLay et al., 2018*). These previous findings prompted us to focus on the mechanisms of *lhx1* expression in regenerating nephric tubules. In order to explore the enhancers that activate *lhx1* gene expression in regenerating nephric tubules, we first searched for evolutionarily conserved noncoding sequences (CNSs) as candidates for the enhancers (*Woolfe et al., 2005*; *Ochi et al., 2012*). In this study, we focused on two categories of CNSs. The first one is evolutionarily conserved between frog and zebrafish, which have a high regenerative capacity, and the other is conserved among human, mice, frog, and zebrafish (*Figure 3*). In order to identify CNSs, we compared the genomic sequence of a 365 kb segment encompassing the human LHX1 gene with the orthologous intervals in mice, opossums, frog (*X. tropicalis*), and zebrafish genomes using the MultiPipMaker alignment tool (*Schwartz et al., 2000*). This analysis identified 20 CNSs conserved between human and fish and 17 CNSs conserved between frog and zebrafish (*Figure 3A*). In order to find the enhancers that genuinely activate gene expression, we used an efficient transgenesis technique for a nonmosaic founder assay of *X. laevis* (*Kroll and Amaya, 1996*). Each *X. tropicalis lhx1*-CNS was cloned into a green fluorescent protein (GFP) reporter plasmid carrying a β-actin basal promoter (*Ogino et al., 2008*). Each construct was used to generate transgenic embryos, and the left side of the nephrons was then resected, after which the embryos were allowed to proceed to stage 37 (*Figure 3B*). Their GFP expression was examined 48 hr after nephrectomy (*Figure 3B*). Reporter constructs without a CNS showed no significant GFP expression (data not shown) (*Ochi et al., 2012*). When GFP expression was detected in the regenerating nephric duct, we counted this as a positive result (*Figure 3C*). CNSs for *lhx1* conserved between frog and zebrafish showed enhancer activities, whereas CNSs conserved among vertebrates showed much stronger enhancer activities (*Figure 3C*). It is known that *lhx1* is expressed in developing nephrons at the early tailbud stage and specifies the renal progenitor cell field (*Taira et al., 1994*; *Carroll et al., 1999*; *Cirio et al., 2011*). Therefore, we examined whether CNSs that have enhancer activities in regenerating nephric tubules are also activated in developing pronephros. The transgenic reporter analysis using early tailbud embryos showed that CNS17, CNS20, and CNS35 have enhancer activities in the eyes and somites at this stage, while lacking (CNS20 and CNS35) or showing (CNS17) very weak enhancer activities in developing pronephros, suggesting that these CNSs primarily function after nephrectomy (*Figure 3—figure supplement 1*). These results suggest that enhancers conserved between human and zebrafish regulate the reactivation of developmental genes in regenerating nephric tubules.

## Arid3a functions as an input transcription factor for *lhx1*-RSREs

The transgenic reporter system of *X. laevis* combined with the surgical removal of nephric tubules revealed that the *lhx1* loci contain the genomic elements that can activate gene expression in regenerating nephrons. We named these elements RSREs. Generally, enhancers activate gene expression by binding transcription factors (*Bulger and Groudine, 2010*). Therefore, the RSREs that we identified must also be regulated by transcription factors. In order to examine the transcription mechanisms for RSREs during nephron regeneration, we first searched for the binding motifs of transcription factors on RSREs. To dissect the transcriptional inputs for the RSREs, the open-access database JASPAR ver. 5 was used to define potential transcription factor binding sites

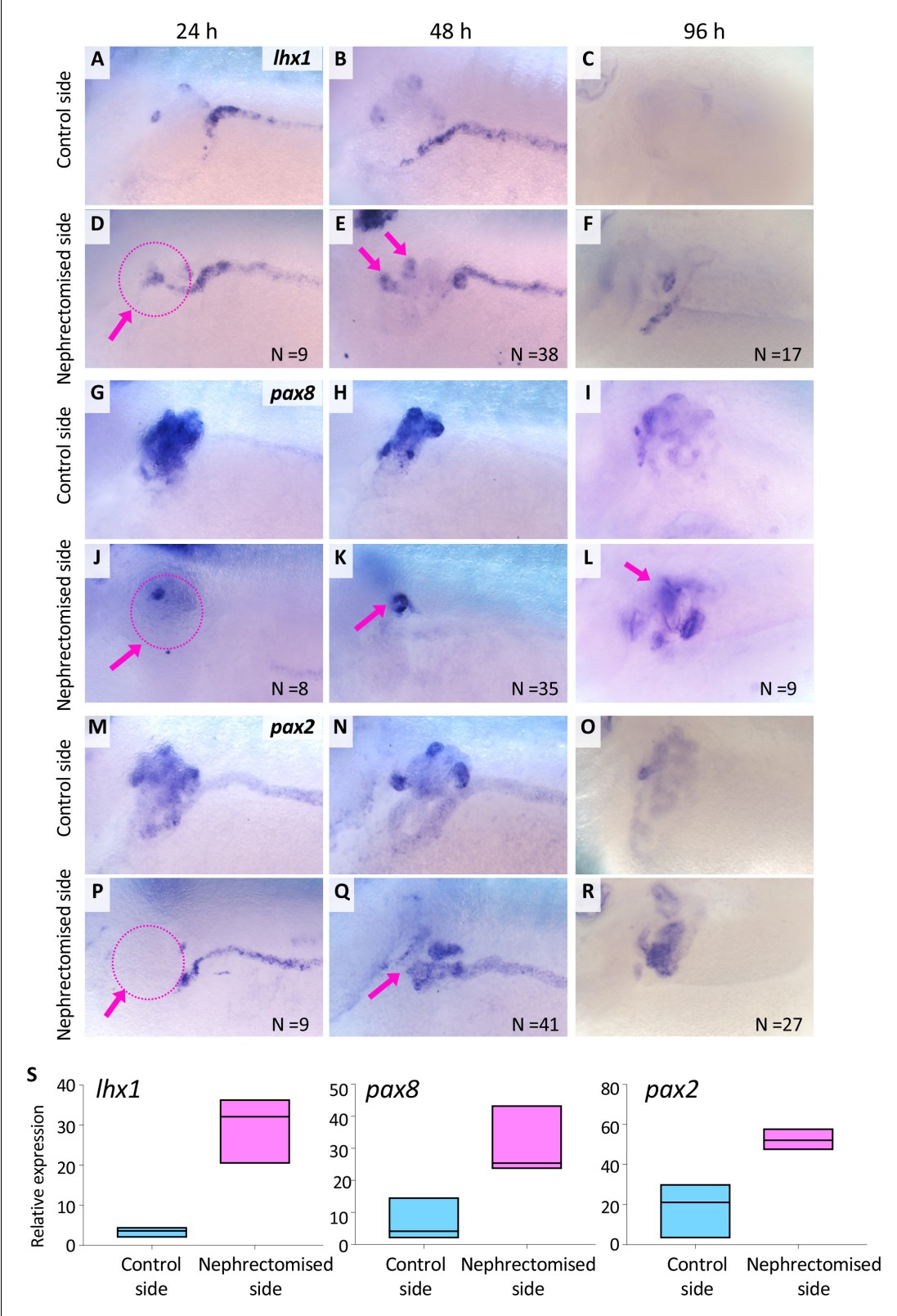

**Figure 2.** *lhx1* expression appears immediately after nephrectomy. (A–F) Expression of *lhx1* on the control side and nephrectomized side at 24, 48, and 96 hr after nephrectomy. N indicates the number of examined embryos. (D) *lhx1* expression appears in regenerating nephric tubules within 24 hr (arrows). (E, F) This expression becomes stronger around 48 hr but disappears at 96 hr. (G–L) Expression of *pax8*. *pax8* expression appears in regenerating nephric tubules and is still detected at 96 hr after nephrectomy. (M–R) Expression of *pax2* is not observed at 24 hr after nephrectomy (P,

*Figure 2 continued on next page*

*Figure 2 continued*

arrow). (**S**) Quantification of expression signals for *lhx1*, *pax8*, and *pax2*. The significance of differences between the control side and the nephrectomized side at 48 hr is calculated by two-tailed paired *t*-test (*lhx1*: *p*=0.0414; *pax8*: *p*=0.0102; *pax2*: *p*=0.0453). Lines in boxes indicate the median.

DOI: https://doi.org/10.7554/eLife.43186.004

The following figure supplement is available for figure 2:

**Figure supplement 1.** Expression of *hnf1b*, *hnf4a*, *osr1*, and *osr2* in regenerating nephric tubules.

DOI: https://doi.org/10.7554/eLife.43186.005

(*Mathelier et al., 2014*). The candidate transcription factors were narrowed down according to their nephric expression using the Expression Atlas (the baseline atlas) (*Petryszak et al., 2014*), and then they were narrowed down further by phylogenetic footprinting (*Figure 4—figure supplement 1*). Although many *lhx1*-CNSs showed enhancer activities in regenerating nephric tubules, we focused on the top three *lhx1* RSREs—CNS17, CNS20, and CNS35—which showed strong enhancer activities. The motif analysis showed that transcription motifs for Arid3a and Spib were commonly found in RSREs (*Figure 4A*). *arid3a* is known to be expressed in the ectoderm of the early neurula and in the epidermis at the late tailbud stage, whereas *spib* is expressed in the anterior ventral blood islands at the neurula stage (*Callery et al., 2005*; *Costa et al., 2008*). We performed an *in situ* hybridization analysis in order to determine whether these genes are also expressed in the nephric tubules and found that *arid3a* is expressed there, whereas *spib* is not (*Figure 4B*, *Figure 4—figure supplement 2*). In addition, we performed immunostaining of Arid3a using *Xla.Tg(Xtr.pax8:EGFP)*, which showed that Arid3a protein was detected in proximal tubules and also in the glomus and/or nephrocoelom (*Figure 4B*, orange arrows and orange arrowheads, respectively). These results suggest that Arid3a is a good candidate for the input transcription factor for RSREs. Therefore, we next examined whether Arid3a regulates the expression of *lhx1* in *Xenopus* and found that *lhx1* was induced by conditionally induced *arid3a* in heat-shock-inducible transgenic *X. laevis* (*Figure 4C*) (*Wheeler et al., 2000*). We then examined whether Arid3a interacts with RSREs in *X. laevis*. The Myc-tagged *X. tropicalis arid3a* mRNA was injected into one-cell-stage *X. laevis* embryos, and then chromatin immunoprecipitation-qPCR (ChIP-qPCR) was performed using anti-Myc antibody. CNS35 was divided into two segments for qPCR, since it is 580 bp long, which is too long for qPCR. ChIP-qPCR showed that Arid3a directly binds to RSREs, but not to exon 5 and CNS32, which do not include putative Arid3a binding motifs (*Figure 4D*). Furthermore, a luciferase reporter analysis using 293 T cells derived from human embryonic kidneys was performed to examine whether Arid3a functions as a transcriptional activator for RSREs. We found that Arid3a activates CNS17 and CNS20 enhancer activities, whereas no activation was observed in the CNS35 reporter (*Figure 4E*). Since CNS35 is located in the intron of *aatf*, it is possible that the activation mechanism of this CNS is different from those of CNS17 and CNS20.

## Arid3a promotes cell cycle progression in regenerating nephrons

During tail regeneration in *X. laevis*, the inhibition of apoptosis results in a failure to induce subsequent cell proliferation (*Tseng et al., 2007*). During proximal tubule regeneration, apoptosis occurs within 3 hr after nephrectomy, and the number of apoptotic cells is decreased at one day (*Caine and Mclaughlin, 2013*). Meanwhile, the nephric tubules begin to extend from the remaining tubules (*Figure 1*). During this regeneration step, cell proliferation must occur, as observed in other types of tissue regeneration (*Poleo et al., 2001*; *Passamaneck and Martindale, 2012*). Therefore, we first examined the time course of cell proliferation after nephrectomy (*Figure 5—figure supplement 1*). Immunofluorescence staining with anti-phosphorylated histone H3 antibody, a marker for mitotic cells, showed that there was no significant difference between the control and the nephrectomized sides until 6 hr after nephrectomy, whereas the number of proliferating cells increased on the nephrectomized side at 24 hr (*Figure 5—figure supplement 1*). It has been shown in a previous study that Arid3a stimulates cyclin E1/E2F-dependent cell cycle progression (*Peeper et al., 2002*). Therefore, we next investigated whether Arid3a contributes to the cell cycle progression. *Xla.Tg(Xla.hsp70:Xtr.arid3a-2A-mcherry, Xtr.pax8:EGFP)* and *Xla.Tg(Xtr.pax8:EGFP)* transgenic *Xenopus* embryos at tailbud stage 23 were treated with or without heat shock and then incubated for 48 hr at

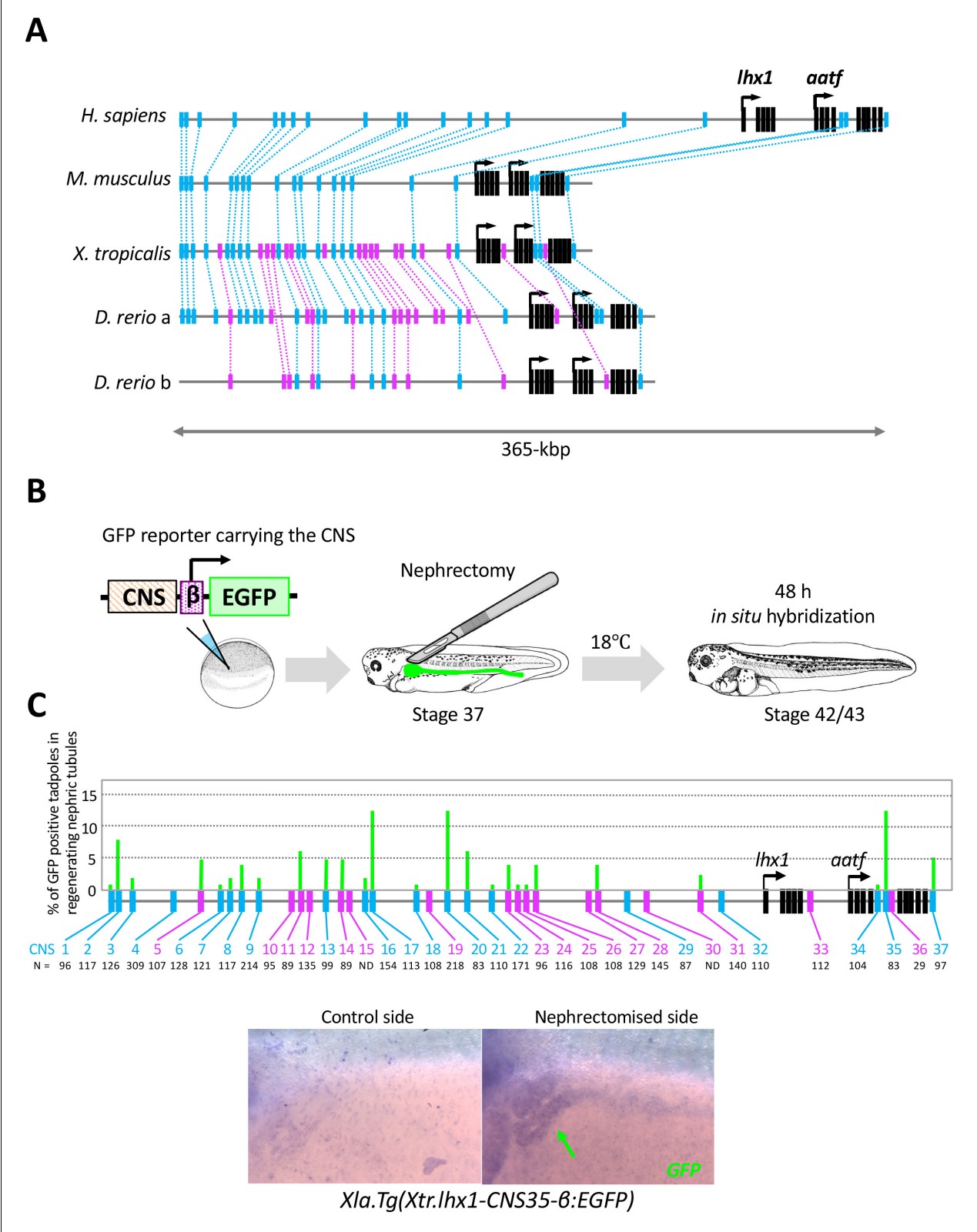

**Figure 3.** The RSRE for *lhx1* is conserved between human and fish. (**A**) A diagram of vertebrate *lhx1* loci showing the position of CNSs. The magenta boxes indicate CNSs conserved between frog and fish, and the blue boxes indicate CNSs conserved between human and fish. The black boxes indicate the exons. (**B**) A diagram of the experimental design for mapping RSREs. GFP reporter constructs carrying *lhx1*-CNSs with the β-actin proximal promoter were subjected to transgenesis. All reporter-injected embryos underwent nephrectomy on the left side at stage 37. Nephrectomized embryos

*Figure 3 continued on next page*

Figure 3 continued
were incubated at 18°C for 48 hr and fixed at stages 42/43. Normally developed embryos were subjected to *in situ* hybridization in order to examine their GFP expression with maximum sensitivity. (C) A summary of RSRE screening. The green bar indicates % of GFP-positive tadpoles in regenerating nephric tubules. N indicates the number of scored tadpoles. The image shows a representative expression pattern of GFP in regenerating nephric tubules. The green arrow indicates the regenerating nephric tubule.

DOI: https://doi.org/10.7554/eLife.43186.006

The following figure supplement is available for figure 3:

**Figure supplement 1.** *lhx1*-CNS17-βEGFP, *lhx1*-CNS29-βEGFP, *lhx1*-CNS35-βEGFP and *pax2*-CNS45-βEGFP were subjected to the transgenic reporter analysis.

DOI: https://doi.org/10.7554/eLife.43186.007

---

18°C. Heat-shock-treated embryos were then sorted by mCherry positivity or negativity (**Figure 5— figure supplement 2**). These tadpoles were examined for the number of phosphorylated histones H3. No significant difference of phosphorylated histones H3 between the heat-shock-untreated embryos and the heat-shock-treated mCherry-negative embryos was observed (**Figure 5—figure supplement 3**). In contrast, the number of phosphorylated histones H3 in the mCherry-positive *Xla.* *Tg(Xla.hsp70:Xtr.arid3a-2A-mcherry, Xtr.pax8:EGFP)* was significantly increased compared with that in the heat-shock-treated *Xla.Tg(Xtr.pax8:EGFP)* and heat-shock-untreated *Xla.Tg(Xla.hsp70:Xtr.* *arid3a-2A-mcherry, Xtr.pax8:EGFP)* (**Figure 5A**). Since we showed that Arid3a can induce *lhx1* expression, we then wondered whether Lhx1 can also promote the cell cycle. Transgenic *X. laevis* in which *lhx1* was conditionally induced showed an increased number of phosphorylated histones H3, as observed in Arid3a-induced *X. laevis* (**Figure 5B**). These results suggest that Arid3a and also Lhx1 promote cell cycle progression after nephrectomy.

## Arid3a with Arid3b and Kdm4a regulate the chromatin configuration of RSREs

The constitutive heterochromatin marker histone 3 lysine nine trimethylation (H3K9me3) is rapidly remodeled to create a chromatin environment, and the lack of H3K9me2/3 could allow global DNA demethylation (**Burton and Torres-Padilla, 2010**). Arid3a is known to stimulate the expression of immunoglobulin heavy chain (IgH) and is also known to change the accessibility of the IgH enhancer (**Kim and Tucker, 2006**; **Lin et al., 2007**). In addition, it has been shown in a previous study that the complex of Arid3a, Arid3b, and Kdm4c modulates the chromatin configuration of stemness genes for breast cancer by decreasing H3K9me3 (**Liao et al., 2016**). Therefore, we next investigated whether Arid3a modulates the H3K9me3 level on RSREs. ChIP-qPCR analysis using anti-H3K9me3 showed that H3K9me3 was enriched not only on the RSREs CNS17 and CNS20, but also on CNS32 and exon 5 of *lhx1* (**Figure 5C**). However, no enrichment of H3K9me3 on CNS35 was observed (**Figure 5C**). Next, we found that the coinjection of *arid3a*, *arid3b*, and *kdm4a* decreased H3K9me3 on the RSREs CNS17 and CNS20, but not on CNS32 (**Figure 5C**). In addition, the enrichment of H3K9me3 on exon 5 was also decreased (**Figure 5C**). Thus, we found that Arid3a directly regulates *lhx1* expression through changing the chromatin configuration on CNS17 and CNS20. In contrast, although CNS35 shows an enhancer activity in regenerating nephric tubules, both the chromatin modification and the activation mechanism differ from those of CNS17 and CNS20.

## Arid3a is required for the regeneration of proximal tubules in *Xenopus*

Our results suggest that Arid3a is involved in the regeneration of nephric tubules in *X. laevis*. In order to investigate whether Arid3a is required for the regeneration of nephric tubules, we applied photo-morpholino oligonucleotide (Photo-MO) that was previously established in zebrafish in order to control the timing of gene knockdown (**Tallafuss et al., 2012**; **Cabochette et al., 2015**). The effectiveness of antisense-MO for gene knockdown is blocked by annealing with sense Photo-MO. Since Photo-MO is cleaved by ultraviolet (UV) treatment, we were able to control the timing of gene knockdown. We first designed the splicing-blocking morpholino for *arid3a.L*, since the expression *arid3a.L* in nephrons is stronger than that of *arid3a.S* (**Figure 4B**, **Figure 4—figure supplement 1A**). The *arid3a.L*-MO-injected embryos became abnormal during gastrulation, consistent with the phenotype that was previously reported using translational blocking morpholino (**Figure 6—figure supplement 1B**) (**Callery et al., 2005**). RT-PCR analysis using control and *arid3a.L*-MO-injected embryos

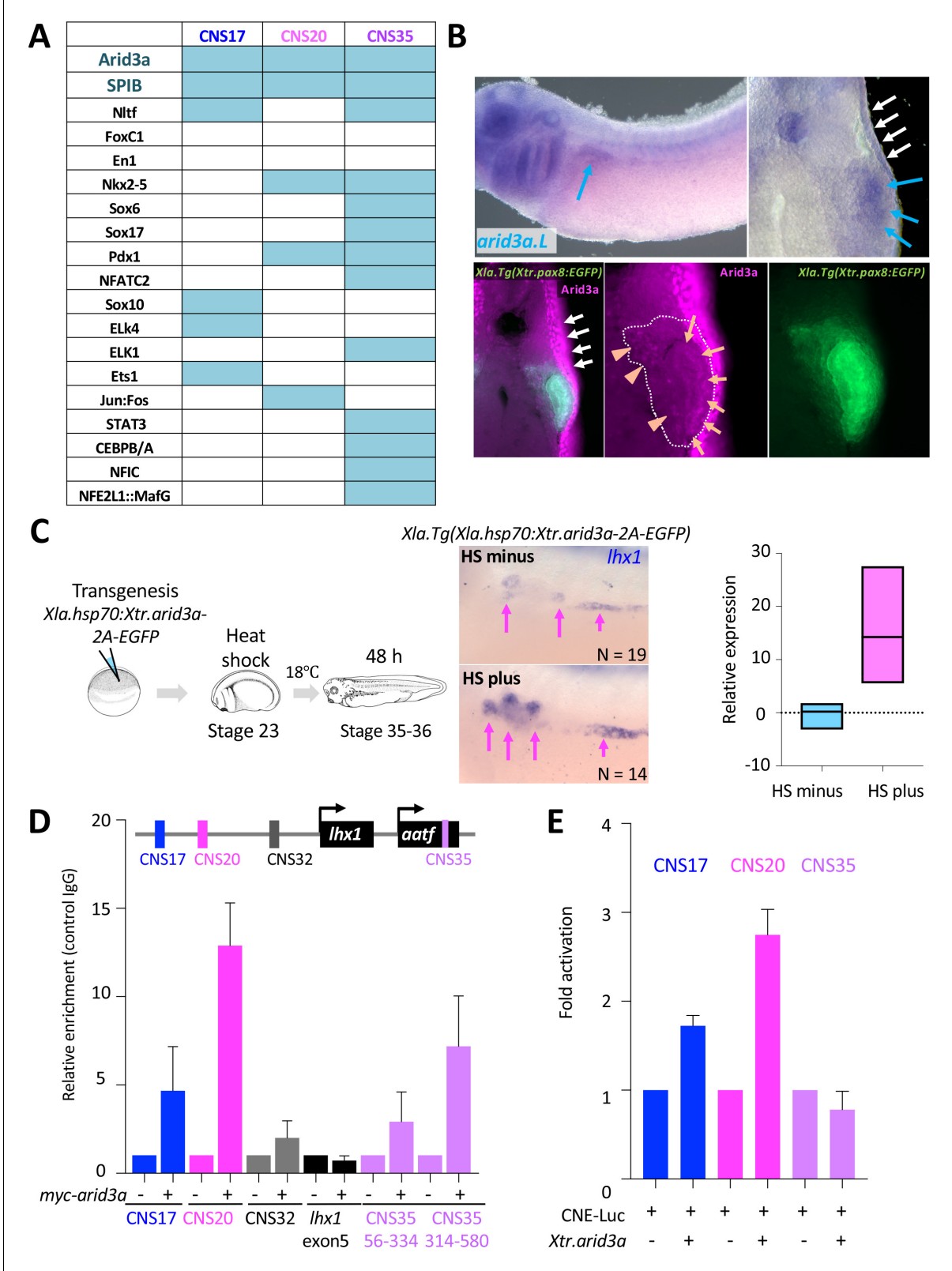

**Figure 4.** Arid3a is an input transcription factor for RSREs. (**A**) A summary of transcription factor binding motifs on RSREs. (**B**) *arid3a* is expressed in nephric tubules. A lateral view of embryos at stage 31 and its transverse section. The upper panels show the *in situ* hybridization of *arid3a.L*. The blue arrows indicate the nephric tubules and the white arrows indicate the epidermis. The lower panels show immunostaining using anti-Arid3a using *Xla.Tg (Xtr.pax8:EGFP)* transgenic tadpole. White arrows: epidermis; orange arrows: proximal tubules; orange arrowheads: glomus and/or nephrocoelom. (**C**)
*Figure 4 continued on next page*

*Figure 4 continued*

Arid3a induces *lhx1* expression. *Xla.Tg(Xla.hsp70:Xtr.arid3a-2A-EGFP)* transgenic *X. laevis* at stage 23 were treated at 34°C for 15 min, followed by 15 min at 14°C. These steps were repeated three times, and tadpoles were incubated at 18°C. *lhx1* expression was observed 48 hr after the heat shock at stages 35–36. The signal intensity of *in situ* hybridization was measured and subjected to statistical analysis. The significance of differences between the control side and the nephrectomized side was calculated using two-tailed unpaired *t*-test (p=0.0131). The magenta arrows indicate *lhx1* expression in proximal and intermediate tubules. N indicates the number of examined embryos. (D) Arid3a directly binds to CNS17, CNS20, and CNS35. Myc-tagged *Xtr.arid3a* mRNA-injected tadpoles were used for ChIP-qPCR. CNS32 and exon 5 were used as negative elements. The significance of differences between the control IgG and the anti-Myc for Arid3a was calculated using two-tailed unpaired Mann–Whitney *t*-test: CNS17, p=0.0286; CNS20, p=0.0022; CNS32, p=0.3143 (not significant); exon5, p=0.7000 (not significant); CNS35 (56-334), p=0.0079; CNS35 (314-580), p=0.0022. The error bars indicate SEM. (E) Arid3a activates CNS17 and CNS20. The luciferase reporter assay was performed using HEK293T cells. The significance of differences between the control vector and the CNS-containing reporter was calculated using two-tailed unpaired *t*-test (CNS17, p=0.0033; CNS20, p=0.0256). The error bars indicate SEM.

DOI: https://doi.org/10.7554/eLife.43186.008

The following figure supplements are available for figure 4:

**Figure supplement 1.** Transcription binding motifs on CNS17, CNS20, and CNS35.

DOI: https://doi.org/10.7554/eLife.43186.009

**Figure supplement 2.** Expression of *arid3a.L*, *arid3a.S*, *spib.L*, and *spib.S* in *X. laevis*.

DOI: https://doi.org/10.7554/eLife.43186.010

showed that the level of transcripts that included exon three was dramatically reduced in *arid3a.L*-MO-injected embryos (*Figure 6—figure supplement 1C*). In contrast, embryos injected with *arid3a.L*-MO annealed with *arid3a.L*-Photo-MO developed normally (*Figure 6—figure supplement 1D*). Thus, *arid3a.L*-MO is rendered inactive by binding to *arid3a.L*-Photo-MO. Next, in order to investigate whether the photosensitive subunit is cleaved by 365 nm light, *arid3a.L*-photo-MO/*arid3a.L*-MO-injected embryos were treated with UV for 30 min at stages 29–30 (*Figure 6A*). RT-PCR analysis showed that the *arid3a.L* transcript, which included exon 3, was significantly decreased by treatment with UV light, same as upon injection of *arid3a.L*-MO alone (*Figure 6A*, *Figure 6—figure supplement 1C*). Since conditionally induced Arid3a promotes the expression of *lhx1*, we wondered whether the expression of *lhx1* in regenerating nephric ducts is affected. *In situ* hybridization analyses showed that there was no significant increase of *lhx1* in UV-treated embryos, suggesting that Arid3a is essential for the expression of *lhx1* after nephrectomy (*Figure 6B*). Cell cycle progression occurred after nephrectomy (*Figure 5—figure supplement 1*). Therefore, we also examined whether *arid3a* knockdown induced by UV affects the cell cycle and found that the number of phosphorylated histone H3-positive cells after nephrectomy in UV-treated embryos was reduced (*Figure 6—figure supplement 2*). We then examined whether Arid3a contributes to the regeneration of proximal tubules. The proximal tubules of *arid3a.L*-photo-MO/*arid3a.L*-MO-injected embryos that were treated with UV were partially removed and subsequently incubated for 72 hr. The regenerated proximal tubules showed a coiled structure in UV-untreated controls (*Figure 6C*). In contrast, although regenerating proximal tubules extended to the front most of the resected *X. laevis* failed to reconstruct the coiled structure, indicating that Arid3a is required for the proper regeneration of nephric ducts in *X. laevis* (*Figure 6C*).

## Discussion

In many cases of tissue regeneration, numerous developmental genes that are evolutionarily conserved among vertebrates become reactivated during regeneration (*Poss, 2010*; *Witman et al., 2011*; *Halasi et al., 2012*; *Shibata et al., 2016*). Therefore, it has been widely discussed that regeneration recapitulates developmental guidance (*Witman et al., 2011*; *Halasi et al., 2012*). Every gene in the genome requires *cis*-regulatory sequences such as a promoter and an enhancer to activate its expression, indicating that genes expressed in regenerating tissues also utilize enhancers and promoters for their expression (*Bulger and Groudine, 2010*). However, no extensive analysis has been performed on whether highly regenerative animals utilize unique *cis*-regulatory elements for regeneration and how such *cis*-regulatory elements become activated during regeneration. Previously, tissue regeneration enhancer elements were identified on the basis of epigenetic profiling (*Kang et al., 2016*). The screening of epigenetic profiles on the basis of open chromatin marks is

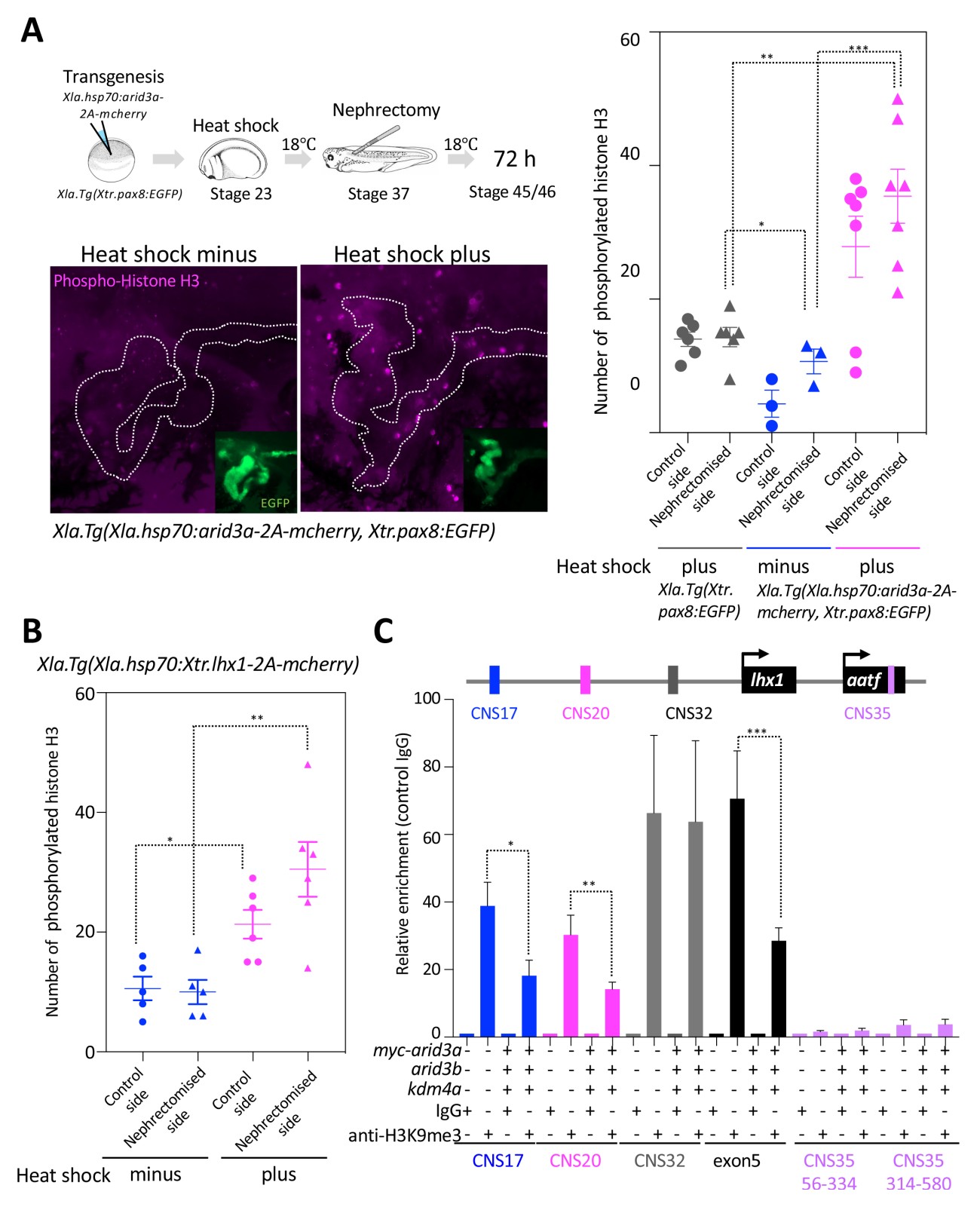

**Figure 5.** Arid3a promotes cell cycle progression. (**A**) The number of phosphorylated histones H3 in the nephrectomized area was increased by the conditionally induced Arid3a. Heat-shocked *Xla.Tg(Xla.hsp70:Xtr.arid3a-2A-mcherry, Xtr.pax8:EGFP)* was nephrectomized at stage 37, incubated for 72 hr, and then fixed at stages 45/46. The white dotted lines indicate the *pax8*-expressing cells, and the magenta indicates the phosphorylated histone H3-positive cells. The left graph shows the statistical analysis; one-way analysis of variance (ANOVA) and Tukey's post hoc test were used. ANOVA

*Figure 5 continued on next page*

*Figure 5 continued*

p=0.0001; *p = 0.5155 (not significant), **p = 0.001, ***p = 0.0019. The error bars indicate SEM. (B) Lhx1 promotes cell cycle progression in the regenerating area. Heat-shocked *Xla.Tg(Xla.hsp70:Xtr.lhx1-2A-mcherry)* was nephrectomized at stage 37 and then incubated for 72 hr. One-way ANOVA and Tukey's post hoc test were used. ANOVA: p=0.0005, *p = 0.0015, and **p = 0.0011. (C) Arid3a with Kdm4a and Arid3b reduced the H3K9me3 levels on RSREs. ChIP analysis was performed using *X. laevis*. Significant differences were calculated by two-tailed unpaired *t*-test. The *p*-values from comparisons between the control and *arid3a*-, *arid3b*-, and *kdm4a*-injected embryos were as follows: *p = 0.0389, **p = 0.0313, and ***p = 0.0456. The error bars indicate SEM.

DOI: https://doi.org/10.7554/eLife.43186.011

The following figure supplements are available for figure 5:

**Figure supplement 1.** Cell cycle progression in regenerating nephric tubules frequently occurs in the remaining proximal tubule and intermediate tubule.

DOI: https://doi.org/10.7554/eLife.43186.012

**Figure supplement 2.** Heat shock induces the expression of mCherry.

DOI: https://doi.org/10.7554/eLife.43186.013

**Figure supplement 3.** Cell cycle progression in heat-shock-untreated and heat-shock-treated tadpoles.

DOI: https://doi.org/10.7554/eLife.43186.014

useful to capture the regulatory blueprint in a genome-wide manner, while our one-by-one transgenic screening based on the actual activities in regenerating tissues provides a novel and powerful approach to directly identify tissue specificity and *in vivo* function. Once we identified the functional enhancers in the regenerating tissues, we were able to search for input transcription factors. Here, our transgenic analysis of functional *cis*-regulatory elements showed that RSREs for *lhx1* are evolutionarily conserved between human and fish, and Arid3a with Arid3b and Kdm4a induces *lhx1* by changing the chromatin configuration of RSREs (*Figure 7A*). In addition, we showed that knockdown of *arid3a* causes the failure of regeneration, whereas the conditionally induced Arid3a promotes cell cycle progression and causes the outgrowth of nephric tubules (*Figure 7B and C*, *Figure 7—figure supplement 1*).

*lhx1* is one of the earliest genes to be expressed in the pronephric anlagen, and its expression disappears during proximal tubule development (*Carroll et al., 1999*; *Carroll and Vize, 1999*) (*Figure 2*). As it recapitulates the developmental program, *lhx1* expression appeared at the early stage in the construction of kidneys *in vivo* (*Takasato et al., 2015*). During *in vivo* nephric regeneration, *lhx1* is one of the earliest genes to be expressed in regenerating nephrons (*Figure 2*). This *lhx1* expression pattern prompted us to examine its noncoding DNA sequences. The *Xenopus lhx1/xlim-1* enhancer responding to activin has been reported to encompass approximately 14.2 kb, including 5 kb upstream and the gene body (*Rebbert and Dawid, 1997*; *Watanabe et al., 2002*). Since recent genome-wide analyses have shown that enhancers are located not only near the gene body but also far away from it, we compared the genomic sequence of a 365 kb segment encompassing the human *LHX1* gene with the orthologous intervals in mice, opossums, *X. tropicalis*, and zebrafish (*Woolfe et al., 2005*; *Ochi et al., 2012*). While the activin-responding *lhx1* enhancer is located in intron 1, enhancers that showed high activities in regenerating nephrons are located 177 kbp (CNS17) and 144 kbp (CNS20) upstream of *lhx1* and are also found in the intron of the next gene *aatf* (CNS35) (*Figure 3*). It is known that many genes have multiple enhancers (*Bulger and Groudine, 2010*). Such multiple enhancers often play different functional roles, while it is also known that they sometimes show overlapping functions (*Osterwalder et al., 2018*). CNS17, CNS20, and CNS35 are activated in regenerating nephric tubules, indicating that these enhancers have overlapping functions. On the other hand, we also found that CNS17 and CNS20 are modified by H3K9me3 but not CNS35 (*Figures 3C* and *5C*). It has been shown in recent studies that functionally redundant enhancers provide phenotypic robustness (*Osterwalder et al., 2018*). Our observation that there are overlapping functions of CNS17, CNS20, and CNS35 in the regenerating nephric tubules, albeit with differential activation mechanisms, suggests that there is a redundancy of *cis*-regulatory elements for regeneration. In addition, we also found that many CNSs that are only conserved between frog and fish showed weak enhancer activities in regenerating nephric tubules (*Figure 3C*). Since these animals have a high regenerative capacity, it is possible that such weak multiple enhancers possessing overlapping activities may also be key to boosting the gene expression in regenerating

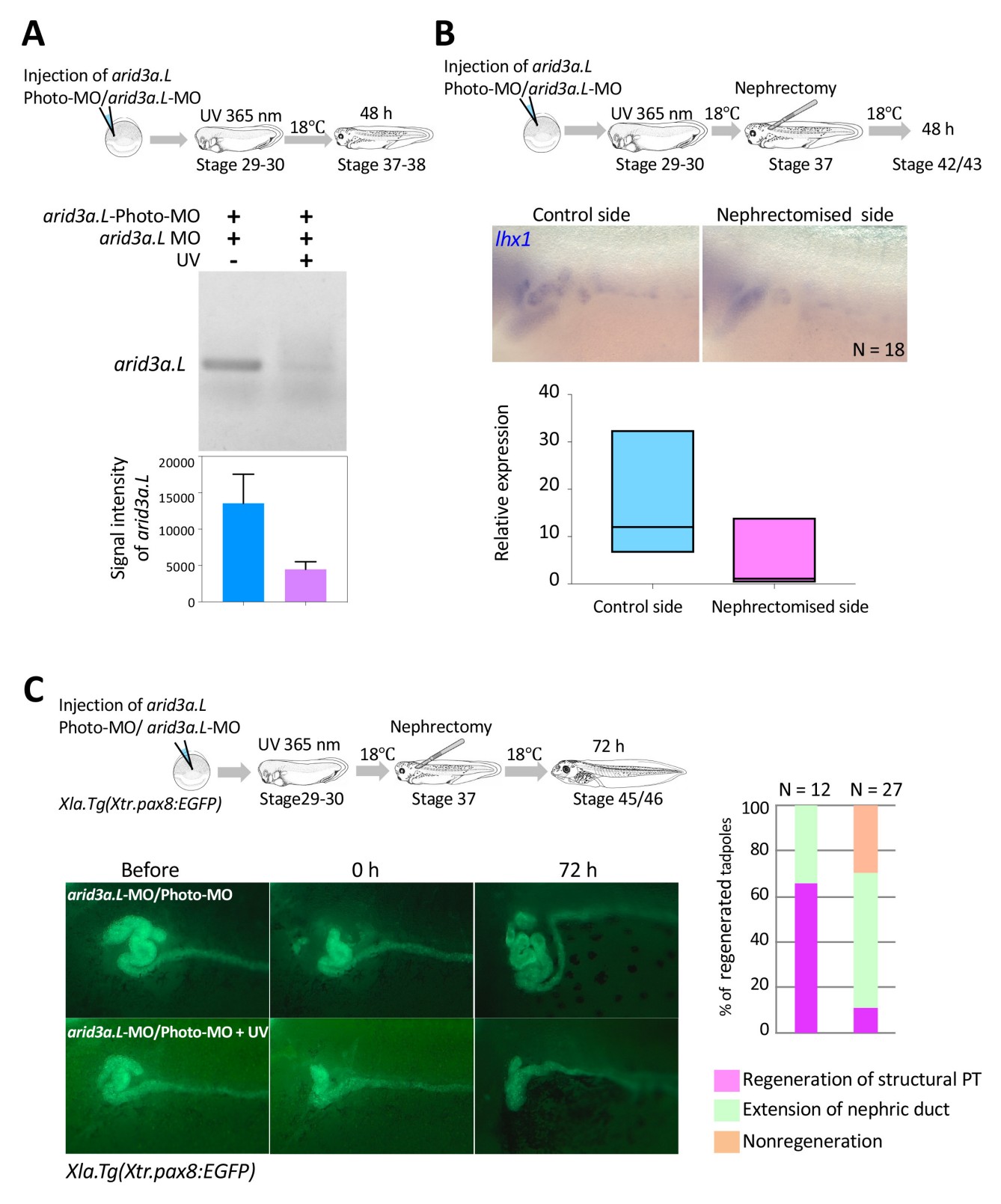

**Figure 6.** Arid3a is required for the regeneration of proximal tubules in *X.laevis*. (**A**) Conditional knockdown of Arid3a using *arid3a.L*-photo-morpholinos (*arid3a.L*-Photo-MO). The upper panel shows the experimental design of conditional gene knockdown experiment using Photo-MO. *arid3a.L*-antisense-splicing-blocking MO (*arid3a.L*-MO) inactivated by Photo-MO is injected at the one-cell stage, subjected to UV exposure at stages 29–30, and then sacrificed for RT-PCR analyses. The lower panel shows the statistical analysis. The significance of differences between the UV-untreated

*Figure 6 continued on next page*

*Figure 6 continued*

and UV-treated embryos was calculated by two-tailed unpaired *t*-test (p=0.0217). The error bars indicate SEM. (B) Conditional knockdown of Arid3a during nephric regeneration causes the reduction of *lhx1* expression. The upper panel shows the experimental design. *arid3a.L*-MO inactivated by Photo-MO is injected at the one-cell stage, followed by UV exposure at stages 29–30, nephrectomy at stages 36–37, and subsequently incubation for 48 hr. The lower panel shows the quantification of *lhx1* expression signals. The analysis indicates that there was no significant difference between the control side and the nephrectomized side (two-tailed paired *t*-test, p=0.0748). N indicates the number of examined embryos. The lines in boxes indicate the median. (C) Arid3a is required for the regeneration of nephric tubules. The upper panel shows the experimental design. *arid3a.L*-MO inactivated by Photo-MO is injected at the one-cell stage, and UV exposure is performed at stages 29–30, followed by nephrectomy at stages 36–37 and subsequently incubation for 72 hr. The left panel shows a summary of the statistics of three independent experiments.
DOI: https://doi.org/10.7554/eLife.43186.015

The following figure supplements are available for figure 6:

**Figure supplement 1.** *Arid3a*-photo-morpholino blocks the effect of *Arid3a*-antisense morpholino.
DOI: https://doi.org/10.7554/eLife.43186.016
**Figure supplement 2.** Conditional knockdown of *Arid3a* during nephric regeneration causes the reduction of cell cycle progression.
DOI: https://doi.org/10.7554/eLife.43186.017

---

tissues. Further studies of transcriptional mechanisms for frog/fish CNSs may provide novel insights into the unique trait of high regenerative capacity in fish and frog.

In zebrafish, *lhx1a* and *six2* mesenchymal cells reconstruct the functional nephrons (*Diep et al., 2011*). In contrast, our live imaging using *Xla.Tg(Xtr.pax8:EGFP)* together with McLaughlin's previous findings suggests that the wound healing and regrowth of existing tubules occur during the regeneration of the nephric tubules of *X. laevis*, since the remaining nephric tubule cells extend toward the glomus (*Caine and Mclaughlin, 2013*). In mammals, mature tubular epithelial cells rapidly lose their brush border and dedifferentiate into mesenchymal-like cells following acute kidney injury, and these cells proliferate to regenerating nephric tubules (*Maeshima et al., 2014*). Therefore, the regenerative mechanisms of the nephric tubules of *X. laevis* appear to be much closer to the regeneration of nephric tubules in mammals rather than the reconstruction of nephrons from stem cells. Further live imaging studies using transgenic animals with markers of dedifferentiating cells and regrowing cells are required in order to reveal the detailed regenerative mechanisms of nephric tubules in vertebrates.

Recently developed *in vivo* organ construction technologies enabled us to make nephric structures derived from iPSCs and ES cells by the combined application of the Wnt activator and other signaling factors such as BMP4 (*Xia et al., 2013*; *Takasato et al., 2014*; *Takasato et al., 2015*). Moreover, direct reprogramming from fibroblasts into renal tubular epithelial cells, called iRECs, has successfully converted mouse and human fibroblasts into renal tubular epithelial cells using a combination of transcription factors: Emx2, Hnf1b, Hnf4a, and Pax8 (*Kaminski et al., 2016*; *Lienkamp, 2016*). In *Xenopus*, it is well known that mRNA injection of Wnt11b, Osr1, Osr2, and Lhx1 with Pax8 can induce ectopic pronephron structures (*Seufert et al., 1999*; *Carroll and Vize, 1999*; *Tételin and Jones, 2010*). Although Arid3a is not strong enough to induce complete ectopic nephrons as with *Osr1*, *Osr2*, *Lhx1*, and *Pax8* mRNA injection, our findings suggest that Arid3a may contribute to the improvement of the efficiency of *in vitro* reconstruction and *in vivo* direct reprogramming or regeneration of nephrons.

For the expression of stem cell factors, Arid3a changes its localization from cytoplasmic to nuclear, which is regulated by importin-9 (*Liao et al., 2016*). It has been shown in a recent study that A disintegrin and metalloproteinase 13 (ADAM13) increases the nuclear localization of a cleavage product of Arid3a and promotes the expression of the target gene tfap2 alpha (*Khedgikar et al., 2017*). ADAMs are known to control a variety of cell functions by modulating the ectodomain shedding of several membrane-anchored signaling molecules (*Blobel, 2005*). In addition, emerging evidence indicates that ADAMs are required for wound healing and the regeneration of oligodendrocytes (*Schönefuß et al., 2012*; *Palazuelos et al., 2015*). Therefore, it is possible that ADAMs change the localization of Arid3a after nephrectomy; nuclear-translocated Arid3a regulates H3K9me3 on RSREs, and this epigenetic alteration triggers *lhx1* expression in regenerating nephric tubules. Further studies are necessary to reveal the molecular basis behind the signal transduction from the extracellular region to the Arid3a complex on *lhx1* RSREs.

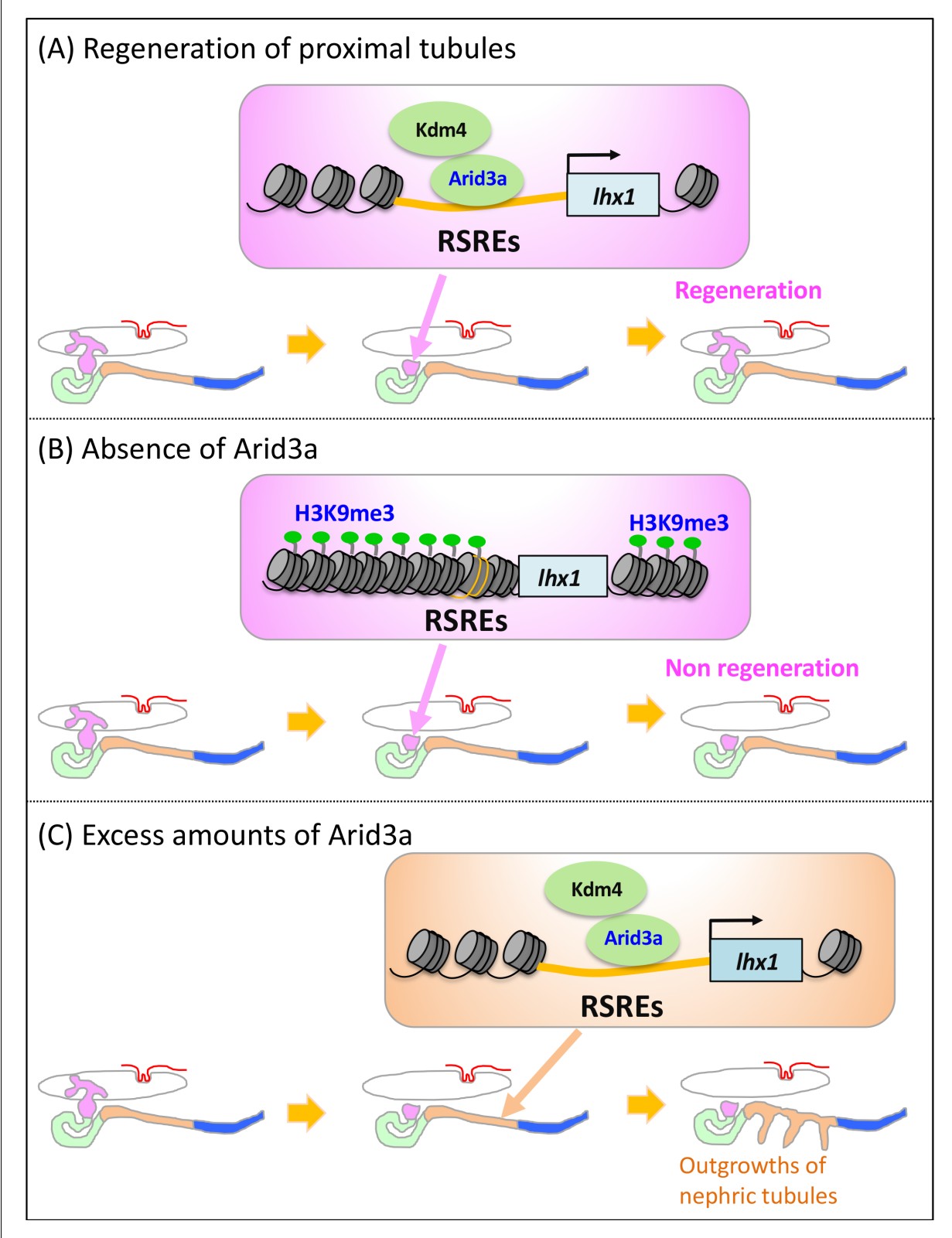

**Figure 7.** Model illustrating the Arid3a function in the regeneration of proximal nephric tubules. (**A**) Arid3a binds to RSREs on *lhx1* and changes the H3K9me3 levels. This chromatin modification allows the induction of *lhx1* expression. (**B**) In the absence of Arid3a, proximal tubules fail to regenerate a complete nephron structure. (**C**) Excess amounts of Arid3a cause the outgrowth of nephric tubules from the distal nephric duct.

DOI: https://doi.org/10.7554/eLife.43186.018

*Figure 7 continued on next page*

*Figure 7 continued*

The following figure supplement is available for figure 7:

**Figure supplement 1.** Conditionally induced *Arid3a* causes the outgrowth of nephric tubules in regenerating nephrons.
DOI: https://doi.org/10.7554/eLife.43186.019

# Materials and methods

## Key resources table

| Reagent type (species) or resource | Designation | Source or reference | Identifiers | Additional information |
|---|---|---|---|---|
| Gene (*Xenopus tropicalis*) | *arid3a* | This paper | RefSeq: NM_001011106.1 | |
| Gene (*Xenopus tropicalis*) | *arid3b* | This paper | RefSeq: XM_002938881.4 | |
| Gene (*Xenopus tropicalis*) | *lhx1* | This paper | RefSeq: NM_001100228.1 | |
| Genetic reagent (*Xenopus laevis*) | *Xla.Tg (Xtr.pax8:EGFP)* | Ochi, H., et al., 2012; doi: 10.1038/ncomms1851. | | |
| Cell line (*Homo sapiens*) | 293T | RIKEN BRC CELL BANK | RCB2202, RRID:SCR_003163 | |
| Transfected construct | pGL4.23 | Promega | E8411 | |
| Transfected construct | pGL-*lhx1*-CNS17-Luc | This paper | | New regent. The CNS17 fragments from IS-*lhx1*-CNS17-β-GFP vector introduced into the SacI and EcoRV sites of pGL4.23 vector. |
| Transfected construct | pGL-*lhx1*-CNS20-Luc | This paper | | New regent. The CNS20 fragments from IS-*lhx1*-CNS17-β-GFP wereintroduced into the SacI and EcoRV sites of pGL4.23 vector. |
| Transfected construct | pGL-*lhx1*-CNS35-Luc | This paper | | New regent. The CNS35 fragments from IS-*lhx1*-CNS17-β-GFP vector were introduced into the SacI and EcoRV sites of pGL4.23 vector. |
| Antibody | Anti-phospho histone H3 (Ser10) antibody | Milipore | 06–570 | (1:1000) |
| Antibody | Anti-Arid3a antibody | DSHB | PCRP-ARID3A-1E9, RRID:AB_2618410 | (1:10) |
| Antibody | Alexa 488 -conjugated goat anti -rabbit IgG | Invitrogen | A11001 | (1:1000) |
| Antibody | Alexa 568 -conjugated goat anti -mouse IgG | Invitrogen, | A11011 | (1:1000) |
| Antibody | Anti-H3K9 (tri-methyl K9) antibody | Abcam, | ab8898 | (1:750) |

*Continued on next page*

*Continued*

| Reagent type (species) or resource | Designation | Source or reference | Identifiers | Additional information |
|---|---|---|---|---|
| Antibody | Mouse monoclonal c-Myc (9E10) antibody | Santa Cruz Bio technology Inc. | sc-40, RRID:AB_291323 | (1:750) |
| Recombinant DNA reagent | *Xenopus laevis* hsp70 promoter | Wheeler, G. N., et al. 2000; doi.org/ 10.1016/S0960 -9822 (00)00596–0 | | |
| Recombinant DNA reagent | IS-β-GFP reporter | Ogino, H., et al., 2008; doi: 10.1242 /dev.009548 | | |
| Recombinant DNA reagent | pCS-myc-arid3a | | | New regent. The PCR product was introduced into the XhoI and XbaI sites of the pCS2 + MT plasmid. |
| Recombinant DNA reagent | pCS-his-arid3b | | | New regent. The PCR product was introduced into the ClaI and XbaI site of the pCS vector. |
| Recombinant DNA reagent | *hsp70-myc-arid 3a-2A-mcherry* | This paper | | New regent. The PCR amplified myc-arid3a and 2A-mcherry were introduced into the ClaI and XbaI sites of the IS-hsp 70-cloning vector. |
| Recombinant DNA reagent | *hsp70-myc-arid 3a-2A-EGFP* | This paper | | New regent. The PCR amplified myc-arid3a and 2A-EGFP were introduced into the ClaI and XbaI sites of the IS-hsp70- cloning vector. |
| Recombinant DNA reagent | *hsp70-lhx1-2A-EGFP* | This paper | | New regent. The PCR amplified *lhx1* was introduced into the EcoRI sites of the pCS2 + MT plasmid. The PCR amplified myc-lhx1 and 2A-EGFP were introduced into the EcoRI and XbaI sites of the IS-hsp70- cloning vector. |
| Recombinant DNA reagent (*Xenopus laevis*) | *arid3a.L* | This paper | Xelaev18006256m | New regent. The PCR product was introduced into the EcoRI and XhoI sites of the pBlue script II SK plasmid. |
| Recombinant DNA reagent (*Xenopus laevis*) | *arid3a.S* | This paper | Xelaev18009788m | New regent. The PCR product was introduced into the EcoRI and XhoI sites of the pBluescript II SK plasmid. |

*Continued on next page*

*Continued*

| Reagent type (species) or resource | Designation | Source or reference | Identifiers | Additional information |
|---|---|---|---|---|
| Recombinant DNA reagent (*Xenopus laevis*) | *spib. L* | This paper | Xelaev18036193m.g | New regent. The PCR product was introduced into the EcoRI and XhoI sites of the pBluescript II SK plasmid. |
| Recombinant DNA reagent (*Xenopus laevis*) | *spib.S* | This paper | Xelaev18037903m.g | New regent. The PCR product was introduced into the EcoRI and XhoI sites of the pBluescript II SK plasmid. |
| Recombinant DNA reagent (*Xenopus laevis*) | *hnf4a* | This paper | Xelaev17043619m, Xelaev17043619m | New regent. The PCR product was introduced into the BamHI and HindIII sites of the pBluescript II SK plasmid. |
| Recombinant DNA reagent (*Xenopus laevis*) | *hnf1b* | This paper | Xelaev18012186m.g, Xelaev18014991m.g | New regent. The PCR product was introduced into the BamHI and HindIII sites of the pBluescript II SK plasmid. |
| Recombinant DNA reagent (*Xenopus laevis*) | *osr1* | This paper | Xelaev14054577m.g, Xelaev14010174m.g | New regent. The PCR product was introduced into the XhoI and BamHI sites of the pBluescript II SK plasmid. |
| Recombinant DNA reagent (*Xenopus laevis*) | *osr2* | This paper | Xelaev14045820m.g, Xelaev14031017m.g | New regent. The PCR product was introduced into the XhoI and BamHI sites of the pBluescript II SK plasmid. |
| Recombinant DNA reagent (*Xenopus laevis*) | *six2* | This paper | Xelaev16000858m, Xelaev16036496m | New regent. The PCR product was introduced into the HindIII and XhoI sites of the pBluescript II SK plasmid. |
| Recombinant DNA reagent (*Xenopus laevis*) | *lhx1* | This paper | Xelaev1604 4871m.g | New regent. The PCR product was introduced into the SmaI and HindIII sites of the pBluescript II SK plasmid. |
| Recombinant DNA reagent (Xenopus laevis) | *pax2* | *Heller and Brändli, 1997*: doi.org/ 10.1016/S0925- 4773(97)00158-5 | | |

*Continued on next page*

*Continued*

| Reagent type (species) or resource | Designation | Source or reference | Identifiers | Additional information |
|---|---|---|---|---|
| Recombinant DNA reagent (Xenopus laevis) | *pax8* | *Heller and Brändli, 1999*: doi.org/ 10.1002/(SICI)1520- 6408(1999) 24:3/4 < 208::AID- DVG4 > 3.0.CO;2 J | | |
| Recombinant DNA reagent (*Mus musculus*) | kdm4a | Mammalian Gene Collection (MGC) Clones | 4207552 | BC028866 |
| Sequence-based reagent (*Xenopus tropicalis*) | PCR primers for CNS | This paper | | |
| Sequence-based reagent (*Xenopus laevis*) | ChIP-qPCR primers | This paper | | |
| Sequence-based reagent (*Xenopus laevis*) | Photo-Morpholino oligonucleotide for *arid3a,L* | This paper | Gene Tools, LLC | AGAGGGAAGCCAG CAGGTACTCACC |
| Sequence-based reagent (*Xenopus laevis*) | Morpholino oligonucleotide for *arid3a,L* | This paper | Gene Tools, LLC | AGTACCTGpT GGCTTCCCT |
| Sequence-based reagent (*Xenopus laevis*) | PT-PCR primers for *arid3a.L* | This paper | | |
| Sequence-based reagent (*Homo sapiens*) | hg19 chr17- 34994909–35360679 | hg19 | UCSC Genome Browser, RRID:SCR_005780 | |
| Sequence-based reagent (*Mus musculus*) | mm10 chr11- 83838963–85151744 | mm10 | UCSC Genome Browser, RRID:SCR_005780 | |
| Sequence-based reagent (*Monodelphis domestica*) | monDom5 chr2- 185210169 –185976291 | monDom5 | UCSC Genome Browser, RRID:SCR_005780 | |
| Sequence-based reagent (*Xenopus tropicalis*) | xenTro3 GL173152 -472286-845619 | xenTro3 | Xenbase, RRID:SCR_003280 | |
| Sequence-based reagent (*Danio rerio*) | danRer10-chr 15_27468859–28180541 | danRer10 | UCSC Genome Browser, RRID:SCR_005780 | |
| Sequence-based reagent (*Danio rerio*) | danRer10- chr5_55422952–55633560 | danRer10 | UCSC Genome Browser, RRID:SCR_005780 | |
| Software, algorithm | GraphPad Prism 7.0 | GraphPad Software | RRID:SCR_002798 | |
| Software, algorithm | Adobe Photoshop | Adobe | RRID:SCR_014199 | |
| Software, algorithm | MultiPipMaker | Schwartz, S., et al., 2000: doi: 10.1101/gr.10.4.577 | RRID:SCR_011806 | |
| Software, algorithm | JASPAR ver. 5 | Mathelier, A., et al., 2014: doi: 10.1093 /nar/gkt997. | RRID:SCR_003030 | |
| Commercial assay or kit | ISOGEN | NIPPON GENE | Code No. 317–02503 | |

*Continued on next page*

*Continued*

| Reagent type (species) or resource | Designation | Source or reference | Identifiers | Additional information |
|---|---|---|---|---|
| Commercial assay or kit | Dual-Luciferase Reporter Assay System | Promega | E1910 | |
| Commercial assay or kit | Dynabeads Protein A | Dynabeads | 10001D | |
| Chemical compound, drug | jetPEI (transfection) | Polyplus-transfection SA | 101–10N | |

## Construction of reporter plasmids

The GFP reporter plasmid carrying a chicken $\beta$-actin basal promoter (−55 to +53) was previously described as $\beta$-GFP (*Ogino and Ochi, 2009*). The CNSs were amplified from *X. tropicalis* genomic DNA by PCR and cloned into $\beta$-GFP reporter vectors. The primer sequences used in this study are summarized in *Supplementary file 1*.

## Identification of CNSs

The 365 kb genomic sequence of the human *LHX1* locus [hg19 chr17-34994909–35360679 (365,771 bp)] and its orthologous sequences in mice [mm10 chr11-83838963–85151744 (1,312,782 bp)], opossums [monDom5 chr2-185210169–185976291 (766,123 bp)], *X. tropicalis* [xenTro3 GL173152-472286-845619 (373,334 bp)], and zebrafish [danRer10-chr15_27468859–28180541 (711,683 bp), danRer10-chr5_55422952–55633560 (210,609 bp)] were downloaded from the UCSC Genome Browser and Xenbase Genome Browser. These sequences were aligned using MultiPipMaker (*Schwartz et al., 2000*).

## Cloning of *X. tropicalis Arid3a* and *Arid3b*

Full-length cDNA fragments of *arid3a* and *arid3b* were amplified from a cDNA pool of *X. tropicalis* tailbud embryos (stage 26). The product was introduced into the XhoI and XbaI sites of the pCS2 +MT plasmid and the ClaI and XbaI site of the pCS2 vector. The PCR-amplified *X. laevis heat-shock promoter*, *Xtr.arid3a*, and *2A-mcherry* DNA fragments were constructed using the InFusion cloning method (*Wheeler et al., 2000*) (Clontech, Palo Alto, CA, USA). For the *in situ* hybridization probes of *X. laevis*, *arid3a.L* and *arid3a.S* were amplified from a cDNA pool of *X. laevis* tailbud embryos (stages 35/36). The primers used for the vector construction are listed in *Table 1*.

## Transgenic reporter assay

*X. laevis* embryos were generated by the sperm nuclear transplantation method with oocyte extracts (*Kroll and Amaya, 1996*). The manipulated embryos were cultured until stage 37, and all normally developed embryos were subjected to *in situ* hybridization in order to examine their GFP expression with maximum sensitivity. All CNS-carrying reporters were tested at least three times. The frequency of GFP expression varied depending on the constructs, but all constructs exhibited a reproducible expression pattern.

## Nephrectomy

We first undertook training in the surgical removal of nephric tubules using stage 37 *Xla.Tg(Xtr. pax8:EGFP)* embryos in accordance with McLaughlin's method (*Caine and Mclaughlin, 2013*). Then, we removed the nephric tubules of *Xla.Tg(Xtr.lhx1-CNSs:EGFP)*. All nephrectomized *X. laevis* were cultured at 18°C until 24, 48, 72, and 120 hr after nephrectomy.

## Motif analysis for transcription factor binding sites

JASPAR ver. 5, an open-access database, was used to search for potential transcription factor binding sites in nephric enhancers (*Mathelier et al., 2014*). The candidate transcription factors were narrowed down according to their expression using the Expression Atlas (*Petryszak et al., 2014*). CNSs

**Table 1.** Primer sequences for the RT-PCR and quantitative RT-PCR.

| | |
|---|---|
| Xtr.arid3a_full-length-F | ATGAAGCTGCAAGCGGTG |
| Xtr.arid3a_full-length-R | TCAGGGAGAAGGATTGTTAG |
| Xtr.arid3b_full-length-F | CGATGCCGCCACCATGCACCATCA CCACCATCATCACCACCATCACT |
| Xtr.arid3b_full-length-R | CTAGAGTGATGGTGGTGATGATGG TGGTGATGGTGCATGGTGGCGGCAT |
| Xla.CNS17-qPCR-F | CTGAGTGAGTTTCAAATAAAAGGATTAAG |
| Xla.CNS17-qPCR-R | GCTATGTAGAGTGGAATAGAGTTAGAATGA |
| Xla.CNS20-qPCR-F | AATACTCACACAGGGAAGACAGC |
| Xla.CNS20-qPCR-R | AAGGCCAAAATTACTTTTCATTTATCTTA |
| Xla.CNS32-qPCR-F | GGGAATTAACCCCCATGGGAA |
| Xla.CNS32-qPCR-F | TTTGCCTCCCTCCTGATCTATAGG |
| Xla.exon5-qPCR-F | CCAGGTTCCATGCACTCTATG |
| Xla.exon5-qPCR-R | TTTCTGGTGGGTGTGACAAA |
| Xla.CNS35-qPCR-56–334 F | AGTTTATAATCTCTGCCGTGCT |
| Xla.CNS35-qPCR-56–334 R | TGTGCTGCTTGGAATTCAAG |
| Xla.CNS35-qPCR-314–580 F | CTTGAATTCCAAGCAGCACAT |
| Xla.CNS35-qPCR-314–580 R | CCTCAAGAACAATTCTCATTTAAATCCAC |
| Xla.arid3a-L-exon2-RT-PCR-F | CCCAAGCAATCTAGTCAACAGACATTCC |
| Xla.arid3a-L-exon4-RT-PCR-R | GCTGCACTGGTGATTGAAGTTGGTAG |
| Xla.lhx1-F | TCTACTGTAAAAACGACTTCTTCAGG |
| Xla.lhx1-R | CCATTGACTGATAGAGAAGAAAAGG |
| Xla.six2.L-F | CGAAGCCAAAGAGAGGTACG |
| Xla.six2.L-R | TTGGGATCCTTCAACTCTGG |
| Xla.six2.S-F | ACCCGTTGTCCTCTTCAATG |
| Xla.six2.S-R | TGACCTGCTGAATGCAAGT |
| Xla.osr1-F | TCCTTCCTACAAGCCTTCAATGGAC |
| Xla.osr1-R | CTGAACAGAACACAATCATGTACAAGGAATTC |
| Xla.osr2-F | GGGAAGATGGGCAGCAAAGCT |
| Xla.osr2-R | TAGAAGTCCTGTCTGGGGCTGTG |
| Xla.hnf1b-F | TGGCTATGGATGCCTATAGTACTGGCC |
| Xla.hnf1b-R | TGCTGATGCTGCTAGTATCTGTGACAAC |
| Xla.hnf4a-F | CGGCTTTCTGTGAACTTCCACTGG |
| Xla.hnf4a-R | CTACATAGCTTCCTGTTTGGTGATGGTC |

DOI: https://doi.org/10.7554/eLife.43186.020

were aligned using ClustalW, and conserved sequences for the candidate transcription factor binding sites were further analyzed by phylogenetic footprinting (*Blanchette and Tompa, 2002*).

### *In situ* Hybridization and Immunostaining

The cDNA clones for *X. laevis lhx1*, *pax8*, *hnf4a*, *hnf1b*, *osr1*, *osr2*, and *six2* probes were synthesized from stages 35/36. The primers used for cDNA cloning are listed in *Table 1*. *pax2* and *pax8* cDNA clones were kind gifts from Dr. Brändli (*Heller and Brändli, 1997*; *Heller and Brändli, 1999*). The nephrectomized transgenic *X. laevis* were subjected to *in situ* hybridization in order to examine their GFP expression with maximum sensitivity. All CNS-carrying reporters were tested at least three times. The frequency of GFP expression varied depending on the constructs, but all constructs exhibited a reproducible expression pattern. The CNSs that drove nephric expression in more than 10% of the examined embryos were defined as RSREs. For whole-mount immunostaining,

nephrectomized *X. laevis* were fixed in 3.7% formaldehyde/MEM at 4°C overnight. After fixation, embryos were washed with 100% ethanol and then incubated in a 2% BSA/PBS-t blocking solution. The embryos were then transferred to a primary antibody solution. A 1 : 1,000-diluted rabbit anti-phospho-histone H3 (Ser10) antibody (Millipore, 06–570) or 1 : 10-diluted anti-Arid3a antibody (DSHB, PCRP-ARID3A-1E9) was used as the primary antibody in conjunction with 1 : 1,000-diluted Alexa 488-conjugated goat anti-rabbit IgG (A11001; Invitrogen) and 1 : 1,000-diluted Alexa 568-conjugated goat anti-mouse IgG (A11011; Invitrogen) secondary antibodies. Images were acquired using ApoTome.2 (Axio Zoom.V16; Carl Zeiss).

## Quantification of *in situ* Hybridization Signals

*In situ* hybridization signals were photographed using an Axio Zoom.V16 with AxioCam MRc cameras (Carl Zeiss). The images were then subjected to further analysis using Adobe Photoshop CS (Adobe, San Jose, CA, USA) as described in the Results. The *in situ* hybridization signals were first inverted, and then the relative luminosity, determined as the signal-positive nephric duct area minus an equal area of background, was measured. Then, the average luminosity was subjected to statistical analysis. The significance of differences between the control side and the nephrectomized side was calculated in Prism7 using two-tailed paired *t*-test (GraphPad Software).

## Quantification of proliferation

The nephrectomized *X. laevis* processed for immunohistochemistry with phosphor-histone H3 (H3P) were photographed at the same magnification and exposure time. H3P-positive cells were counted in a square region (160 mm$^2$) of regenerating tubules indicated by GFP expression of *Xla.Tg(Xtr. pax8:EGFP)*.

## Luciferase reporter assay

For the luciferase reporter assay, *lhx1*-CNSs were cloned into the pGL4.23 vector (Promega, Madison, WI, USA). HEK293T cells seeded in 24-well plates were transfected with 100 ng of *lhx1*-CNS luciferase plasmids and 10 ng of *Renilla* luciferase plasmids using jetPEI (Polyplus Transfection SA, Illkirch, France). The total amount of DNA per well was adjusted to 1 μg with pBKS plasmids. Transfected cells were incubated for 48 hr, and then the luminescence signals were measured following the manufacturer's protocol (Promega).

## ChIP analysis for *lhx1*-CNSs

ChIP was performed as described previously (*Gazdag et al., 2016*). *Myc-Xtr.arid3a-*, *Xtr.Arid3b-*, and mouse *kdm4a* mRNA-injected embryos were collected at stages 35/36 and subjected to ChIP using Dynabeads Protein A (Dynabeads, Great Neck, NY, USA). The following antibodies were used: rabbit polyclonal anti-H3K9 (tri-methyl K9) antibody (ab8898; Abcam, Cambridge, MA, USA). and mouse monoclonal c-Myc (9E10) antibody (sc-40; Santa Cruz Biotechnology Inc., Santa Cruz, CA, USA). Decrosslinking and elution of DNA were performed for real-time quantitative PCR using StepOne Plus (Thermo Fisher Scientific, Waltham, MA, USA). Injection and ChIP-qPCR were performed at least three times independently.

## Conditional knockdown using Photo-Morpholino

The splicing-blocking antisense-MOs for *Arid3a.L* and photo-morpholinos (Gene Tools, Philomath, OR, USA) used in this study were as follows:

- *arid3a.L*-exon2-intron2-MO: AGAGGGAAGCCAGCAGGTACTCACC.
- *arid3a.L*-exon2-intron2-photo-MO: AGTACCTGpTGGCTTCCCT.

A ratio of 0.9 : 1 of photo-MO to MO was annealed at room temperature for 30 min, and then 2 nM photo-MO/MO was injected with a GFP tracer into one-cell-stage embryos. Cleavage of the antisense of photo-MO was performed by exposing embryos to UV light (365 nm) for 30 min.

## Acknowledgments

This work was supported by Grants-in-Aid for Scientific Research from the Japan Society for the Promotion of Science (JSPS) (Grants nos. 16K07362, 25124704, 25125716, 16H04828, and 17KT0049 to

H Ochi and 15K07082 and 16H04794 to H Ogino), as well as grants from CREST (JST to H Ogino), the Takeda Science Foundation (to H Ochi and H Ogino), Suzuken Memorial Foundation, and YU-COE (C) (H Ochi). The genomic DNA of *X. tropicalis* for the screening enhancer was provided by the Amphibian Research Center, Hiroshima University, with partial support from the National Bio-Resource Project of AMED, Japan.

## Additional information

### Funding

| Funder | Grant reference number | Author |
|---|---|---|
| Japan Society for the Promotion of Science | 16K07362 | Haruki Ochi |
| Takeda Science Foundation | | Haruki Ochi<br>Hajime Ogino |
| Suzuken Memorial Foundation | | Haruki Ochi |
| Japan Society for the Promotion of Science | 15K07082 | Hajime Ogino |
| Japan Society for the Promotion of Science | 16H04794 | Hajime Ogino |
| Japan Society for the Promotion of Science | 25124704 | Haruki Ochi |
| Japan Society for the Promotion of Science | 25125716 | Haruki Ochi |
| Japan Society for the Promotion of Science | 16H04828 | Haruki Ochi |
| Japan Society for the Promotion of Science | 17KT0049 | Haruki Ochi |

The funders had no role in study design, data collection and interpretation, or the decision to submit the work for publication.

### Author contributions

Nanoka Suzuki, Data curation, Formal analysis, Investigation; Kodai Hirano, Investigation, Construction of plasmids; Hajime Ogino, Resources, Funding acquisition, Methodology, Construction of plasmids; Haruki Ochi, Conceptualization, Data curation, Formal analysis, Supervision, Funding acquisition, Validation, Investigation, Writing—original draft, Project administration, Writing—review and editing

### Author ORCIDs

Haruki Ochi http://orcid.org/0000-0002-0088-581X

### Ethics

Animal experimentation: This study was performed in strict accordance with the recommendations in the Act on Welfare and Management of Animals in Japan and the Guide for the Care and Use of Laboratory Animals of the National Institutes of Health. All of the animals were handled in accordance with the Animal Research Guidelines at Yamagata University. The protocol was approved by the Committee on the Ethics of Animal Experiments of Yamagata University.

### Decision letter and Author response

Decision letter https://doi.org/10.7554/eLife.43186.024
Author response https://doi.org/10.7554/eLife.43186.025

# Additional files

## Supplementary files

• Supplementary file 1. The primer sequences used in this study.
DOI: https://doi.org/10.7554/eLife.43186.021

• Transparent reporting form
DOI: https://doi.org/10.7554/eLife.43186.022

## Data availability

All data were generated and analysed during this study are included in the manuscript and supporting files.

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
