## [Decision Letter]

[Editors’ note: a previous version of this study was rejected after peer review, but the authors submitted for reconsideration. The first decision letter after peer review is shown below.]

Thank you for submitting your work entitled "Arid3a regulates the nephric tubule regeneration via the evolutionarily conserved regeneration signal-response enhancer" for consideration by *eLife*. Your article has been reviewed by three peer reviewers, and the evaluation has been overseen by a Senior/Reviewing Editor. The reviewers have opted to remain anonymous.

Our decision has been reached after consultation between the reviewers. Based on these discussions and the individual reviews below, we regret to inform you that your work will not be considered further for publication in *eLife*.

As you will see from the reviews below, the reviewers have raised several major issues that would need to be addressed before the manuscript could be considered further:

1) You would need to provide evidence that the regeneration specific enhancers indeed are regeneration specific.

2) You would need to provide evidence that Arid3a is expressed in the pronephros.

3) You would need to provide a negative control for the chromatin immunopprecipitation experiments.

I refer you to the full reviews for further details. If some time in the future, you feel you can provide the necessary data, we would be willing to consider new version of the manuscript that contains these new data and every effort would be made to return the manuscript to the same reviewers.

*Reviewer #1:*

The manuscript by Suzuki et al. describes how Arid3a regulates enhancers of *lhx1* gene expression during *Xenopus*nephron regeneration.

Previous studies have demonstrated that the *Xenopus*nephron regenerates after partial nephrectomy and that *lhx1* gene activation occurs during nephron regeneration in zebrafish. This study indicates that *lhx1* enhancer elements enable binding of Arid3a to activate of *lhx1* gene expression during nephron regeneration. It also shows that Arid3a recruits chromatin modifiers to these enhancer elements to facilitate *lhx1* gene expression and resultant cell proliferation. The topic of the paper is significant in that it is the first paper to indicate a mechanism by which regeneration of nephrons occurs in *Xenopus*. More broadly, this paper is the first to examine how enhancers facilitate regeneration of the nephron. This paper is appropriate for publication in *eLife* if the following concerns are addressed in a revised manuscript.

1) The manuscript focuses on Arid3a association with a regeneration response enhancer of *lhx1* (and to a certain extent *pax8*), but the Introduction fails to describe the role of *lhx1* (and *pax8*) in *Xenopus* development or cite manuscripts related to it known roles in kidney development (Taira et al., 1994; Carroll and Vizeet al., 1999; Carroll, Wallingford and Vize, et al.1999; DeLay et al., 2018).

2) Results related to Figure 1 do not describe when (what stage) the nephrectomy is performed in *this* study.

3) Because *pax8* is one of the first genes expressed during kidney development, the *pax8* transgenic line is an excellent choice to examine regeneration. However, since in Figure 2 there is some regenerating tissue that is labeled by *lhx1* probes but not *pax8* probes, is it possible that the transgenic *pax8* line does not label all of the regenerating tissue? If so, this is useful information in the context of Figure 1.

4) Throughout the manuscript it is unclear whether these enhancer elements are important for *lhx1* expression during normal development or just during regeneration.

5) In Figure 3C, the image of the control side is not clear as to whether the conserved noncoding sequence drives GFP expression in the nephron in the absence of nephrectomy.

6) In Figure 4B, kidney expression of Arid3a.L is difficult to see in the images.

7) The explanation of Figure 5A is unclear. How does inset panel relate to position of larger image? What does the purple indicate?

*Reviewer #2:*

This paper identifies enhancers that are used in regeneration of the kidney to drive *lhx1* expression. As an important part of regeneration, this is potentially very interesting. However, I don't see experiments addressing whether the regeneration induced enhancers are also normally used in development? What is the implication if they are also active in Development – does it mean that the enhancer is not specific to regeneration? The authors call the enhancers regeneration specific, so this is a crucial point for the paper as presented. While this may be the main point that needs to be addressed in a revision, there are a number of other points that need to be clarified in the paper.

Arid3a binding and function:

Figure 4B: the expression of Arid 3aL in tubules is very hard to see. Perhaps a higher magnification would help.

Figure 4C: are some of these tadpoles nephrectomized as suggested in the legend? I don't see panels that address this.

Figure 4C: is CNE35 a negative control? It also binds Arid3a, but this putative enhancer is not activated in expression, so what is the implication here? The possibility that it only activates the gene in which it is resident is attractive but not tested explicitly, though such a specific context effect would be interesting. In any case, such ChIP experiments should address binding to a negative control element or two. As presented, the data are consistent with general binding of the tagged Arid3a to chromatin in a nonspecific way.

I am confused about the Arid3a expression, according to Xenbase the levels of expression in kidney are similar for Arid3aL and 3aS, and in the figures presented, neither appears to be expressed in the pronephric region, though Arid3aL has an overall higher staining in the ISH than does Arid3aS. However, this is not as well controlled as the RNA seq in Xenbase, though that is for adult kidney.

The logic of the experiments in Figure 6 would be clearer if it were made clear that the photoMO is the sense strand which binds and inhibits the antisense.

The results with the photoactivable MO appear quite convincing, though I am still puzzled by how it targets the pronephros, when the evidence for *arid3aL* expression is not so clear.

*Reviewer #3:*

This paper presents a thorough and interesting analysis of conserved non-coding elements in the *lhx1* locus. This is significant since *lhx1* expression and function is important in kidney development and regeneration. The authors show that CNEs have sites for Arid3a and go on to show that Arid3a can bind and activate *lhx1*CNEs. They also show that a consequence of Arid3a regulation of *lhx1* is increased proliferation. This work will be of interest to kidney development investigators and to labs studying tubule regeneration. The approach is rigorous in that many replicates were performed for each experiment and statistical methods were applied.

1) What distinguishes the Regeneration Signal Responsive Enhancers described in this work from developmental regulatory elements in the *lhx1* locus? i.e. are these elements the same as *cis*-elements active during development? Since the fish to human enhancer conservation was stronger than the fish to frog enhancer conservation, this seems to suggest the RSREs might simply be developmental enhancers reactivated by injury, raising the question of whether they warrant the name "regeneration signal response enhancers". Also, wouldn't it have been more interesting to look at the CNEs that were specific to fish and frog (which can regenerate), even if they were not as strong, and look for binding sites in those that were not present in mouse/human (which don't regenerate as well)?

2) As described by the authors, the rationale for choosing to focus on *lhx1a* as a regeneration gene conflates two different repair mechanisms in frog and fish when the authors say "Among these, the previous finding that *lhx1a*-expressing cells regenerate functional nephrons in zebrafish prompted us to focus on the mechanisms of Lhx1 expression in regenerating [frog] nephric tubules (Diep et al., 2011)." The zebrafish repair mechanism involves Six2-expressing adult kidney stem cells while the frog nephrectomy model appears to be a regrowth of an existing tubule, much like mammalian kidney regeneration/wound healing, and this tubule does not express Six2. It is important to be clear about what each model represents in terms of distinct regenerative mechanisms.

3) Have the authors tried other unbiased approaches to identifying proteins bound to RSREs?

4) In Figure 4D, there is no explanation of why CNE35 was split into 56-334 and 314-580.

5) In Figure 5C why would CNE35 act differently from CNE17 and CNE20 in terms of altered chromatin structure?

6) From the data shown I am not convinced Arid3a is expressed in the pronephros. The magenta arrows in Figure 4B seem to be pointing at yolk with no blue *in situ* reaction product where the pronephros would be. The cross section is too low resolution to see tubule morphology in the stained cross section. Clearer data should be presented.

7) In Figure 4C, the experiment is a little difficult to understand. How many hours after heat shock was *lhx1* expression assayed?

8) Figure 6: What accounts for the loss of message in UV treated photo MO condition? Splice blocking MOs usually alter splicing; is there an altered Arid3a mRNA after UV activation of the MO? Is there a control for UV exposure alone on mRNA abundance? 30 minutes of continuous exposure to 365 nm light could have effects of its own. Also the experimental design at the top of 6A should indicate that both photo and regular MO's were injected together.

---

## [Author Response]

[Editors’ note: the author responses to the first round of peer review follow.]

We performed transgenic experiments on the regeneration signal-response enhancer (RSRE) in embryos at the early tailbud stage in order to examine whether RSREs function in developing nephrons. We observed no enhancer activities in developing nephrons at stage 26. The results suggest that the enhancers that we identified on the basis of the activities in regenerating nephric tubules exert their primary function after nephrectomy to induce *lhx1*, and they are now shown in Fig. 3—figure supplement 1. In order to improve the images of *arid3a* expression, we again performed *in situ* hybridization and also performed immunostaining using anti-Arid3a antibody. To provide a negative-control ChIP-qPCR for anti-Myc antibody, we added two additional genomic elements, *exon5* and CNS32, which do not contain any putative Arid3a binding motifs. We also performed ChIP-qPCR targeting exon5 and CNS32 using antiH3K9me3 antibody.

As you will see from the reviews below, the reviewers have raised several major issues that would need to be addressed before the manuscript could be considered further:1) You would need to provide evidence that the regeneration specific enhancers indeed are regeneration specific.

We performed transgenic experiments of RSREs in embryos at the early tailbud stage in order to examine whether RSREs function in developing nephrons. This transgenic reporter analysis showed that RSREs exhibited no enhancer activities in developing nephrons at stage 26, suggesting that the enhancers that we identified on the basis of the activities in regenerating nephric tubules exert their primary function after nephrectomy to induce *lhx1*.

2) You would need to provide evidence that Arid3a is expressed in the pronephros.

We again performed *in situ* hybridization and provided better images of this. We also performed immunostaining using anti-Arid3a antibody and found Arid3a protein in the proximal tubules.

3) You would need to provide a negative control for the chromatin immunopprecipitation experiments.

We performed ChIP-qPCR experiments using two additional genomic elements, exon 5 and CNS32, which do not contain any putative Arid3a binding motifs, as a negative control for anti-Myc antibody. We also performed ChIP-qPCR targeting exon 5 and CNS32 using antiH3K9me3 antibody.

I refer you to the full reviews for further details. If some time in the future, you feel you can provide the necessary data, we would be willing to consider new version of the manuscript that contains these new data and every effort would be made to return the manuscript to the same reviewers.Reviewer #1:[…] 1) The manuscript focuses on Arid3a association with a regeneration response enhancer of lhx1 (and to a certain extent pax8), but the Introduction fails to describe the role of lhx1 (and pax8) in Xenopus development or cite manuscripts related to it known roles in kidney development (Taira et al., 1994; Carroll and Vize, 1999; Carroll, Wallingford and Vize, 1999; DeLay et al., 2018).

Thank you for this comment. We have included the following text in the “Results” section: “It is known that *lhx1* is expressed in developing nephrons at the early tailbud stage and specifies the renal progenitor cell field (Taira et al., 1994; Carroll et al., 1999; Cirio et al., 2011)”. We have also included the following text in the “Results” section: “It has been shown in a previous study involving simple mRNA injection using *Xenopus* embryos that Lhx1 with Pax8 can induce ectopic pronephron structures (Carroll and Vize, 1999). […] These previous findings prompted us to focus on the mechanisms of *lhx1* expression in regenerating nephric tubules”.

2) Results related to Figure 1 do not describe when (what stage) the nephrectomy is performed in this study.

Thank you for this comment. We have included the stage when nephrectomy was performed in the “Results” section and also in the figures.

3) Because pax8 is one of the first genes expressed during kidney development, the pax8 transgenic line is an excellent choice to examine regeneration. However, since in Figure 2 there is some regenerating tissue that is labeled by lhx1 probes but not pax8 probes, is it possible that the transgenic pax8 line does not label all of the regenerating tissue? If so, this is useful information in the context of Figure 1.

We agree that *Xla.Tg(Xtr.pax:8:EGFP)* does not label all of the regenerating nephric cells. Ideally, a double transgenic line such as *pax8:EGFP* and *lhx1:mechrry* would be required to trace all of the regenerating cells. In order to address this point, we have included the following changes: “Although it is possible that primary induced *lhx1-, pax8-, hnf4a-, hnf1b-,* and *osr2*expressing cells are not completely consistent with cells observed in *Xla.Tg(Xtr.pax8:EGFP)* regenerating nephric tubules, the immediate induction of *lhx1, pax8, hnf4a, hnf1b,* and *osr2* after nephrectomy suggests that the loci of these genes contain enhancers of injury response”.

4) Throughout the manuscript it is unclear whether these enhancer elements are important for lhx1 expression during normal development or just during regeneration.

We agree with this comment. Hence, in response to this point, we have examined the enhancer activities of RSREs in embryos at the early tailbud stage. This transgenic reporter analysis showed that CNS20 and CNS35 do not exhibit enhancer activities in developing nephrons at stage 26, at which endogenous *lhx1* expression has already occurred. Although some of the CNS17 transgenic reporter embryos showed GFP expression in developing pronephros, most of the TG embryos did not. These results suggest that the primary function of RSREs is to induce *lhx1* after nephrectomy (—figure supplement 1Figure 3).

5) In Figure 3C, the image of the control side is not clear as to whether the conserved noncoding sequence drives GFP expression in the nephron in the absence of nephrectomy.

Thank you for pointing that out. We have provided a better image of GFP expression.

6) In Figure 4B, kidney expression of Arid3a.L is difficult to see in the images.

Thank you for pointing that out. We performed the *in situ* hybridization again and provided better images. We also performed immunostaining using anti-Arid3a antibody and found that Arid3a protein was present in the proximal tubule. We have amended the text describing this as follows: “*arid3a* is known to be expressed in the ectoderm of the early neurula and in the epidermis at the late tailbud stage, whereas *spib* is expressed in the anterior ventral blood islands at the neurula stage (Callery et al., 2005; Costa et al., 2008). […] In addition, we performed immunostaining of Arid3a using *Xla.Tg(Xtr.pax8:EGFP)*, whichshowed that Arid3a protein was detected in proximal tubules and also in the glomus and/or nephrocoelom (Figure 4B, orange arrows and orange arrowheads, respectively)”.

7) The explanation of Figure 5A is unclear. How does inset panel relate to position of larger image? What does the purple indicate?

Thank you for this comment. We have amended the figure’s legend as follows: “Heat-shocked *Xla.Tg(Xla.hsp70:Xtr.arid3a-2A-mcherry, Xtr.pax8:EGFP)* was nephrectomized at stage 37, incubated for 72 h, and then fixed at stages 45/46. The white dotted lines indicate the *pax8*expressing cells, and the magenta indicates the phosphorylated histone H3-positive cells”.

Reviewer #2:This paper identifies enhancers that are used in regeneration of the kidney to drive lhx1 expression. As an important part of regeneration, this is potentially very interesting. However, I don't see experiments addressing whether the regeneration induced enhancers are also normally used in development? What is the implication if they are also active in Development – does it mean that the enhancer is not specific to regeneration? The authors call the enhancers regeneration specific, so this is a crucial point for the paper as presented. While this may be the main point that needs to be addressed in a revision, there are a number of other points that need to be clarified in the paper.

Thank you very much for the thoughtful comments. In order to address this main comment, we examined the enhancer activities of RSREs in embryos at the early tailbud stage. This transgenic reporter analysis showed that the RSREs CNS20 and CNS35 have no enhancer activities in developing pronephros. Although some of the CNS17 transgenic reporter embryos showed GFP expression in developing pronephros, most did not. These results suggest that the enhancers that we identified on the basis of the activities in regenerating nephric tubules exert their primary function after nephrectomy to induce *lhx1*, although it is still difficult to determine definitively whether RSREs are specific for regeneration (Figure 3—figure supplement 1). Therefore, we used the term “regeneration signal-response enhancer” rather than “regeneration-specific enhancer” in our manuscript.

Arid3a binding and function:Figure 4B: the expression of Arid 3aL in tubules is very hard to see. Perhaps a higher magnification would help.

Thank you for this comment. We performed *in situ* hybridization again and provided better images. We also performed immunostaining using anti-Arid3a antibody and found that Arid3a protein was present in the proximal tubules. We have amended the part of the manuscript describing this as follows: “*arid3a* is known to be expressed in the ectoderm of the early neurula and in the epidermis at the late tailbud stage, whereas *spib* is expressed in the anterior ventral blood islands at the neurula stage (Callery et al., 2005; Costa et al., 2008). […] In addition, we performed immunostaining of Arid3a using *Xla.Tg(Xtr.pax8:EGFP)*, whichshowed that Arid3a protein was detected in proximal tubules and also in the glomus and/or nephrocoelom (Figure 4B, orange arrows and orange arrowheads, respectively)”.

Figure 4C: are some of these tadpoles nephrectomized as suggested in the legend? I don't see panels that address this.

Thank you for this comment. We have amended this figure’s legend as follows:

“*Xla.Tg(Xla.hsp70:Xtr.arid3a-2A-EGFP)* transgenic *X. laevis* at stage 23were treated at 34°C for 15 min, followed by 15 min at 14°C. These steps were repeated three times, and tadpoles were incubated at 18°C. *lhx1* expression was observed 48 h after the heat shock at stages 35-36”. We have also added a panel to explain the experimental design.

Figure 4C: is CNE35 a negative control? It also binds Arid3a, but this putative enhancer is not activated in expression, so what is the implication here? The possibility that it only activates the gene in which it is resident is attractive but not tested explicitly, though such a specific context effect would be interesting. In any case, such ChIP experiments should address binding to a negative control element or two. As presented, the data are consistent with general binding of the tagged Arid3a to chromatin in a nonspecific way.

Thank you for this comment. We have provided negative-control elements, exon 5 and CNS32, which do not contain putative Arid3a binding motifs. ChIP-qPCR analysis showed that no enrichment of Myc-tagged Arid3a was observed on exon 5 and CNS32.

As the reviewer mentioned, CNS35 was identified as an enhancer that is activated in regenerating nephric tubules, and ChIP analysis showed that Arid3a binds to this CNS, whereas Arid3a failed to activate this enhancer when Arid3a and a luciferase reporter were cotransfected into 293T cells. In addition, we also found that the enrichment of H3K9me3 on CNS35 was not detected. As the reviewer pointed out, this lack of a repression mark suggests that Arid3a stays on CNS35 in the naïve genome and that this state allows the immediate induction of *lhx1.* However, we could not test this attractive possibility in this study. Instead, we included the following sentences in the “Discussion” section: “Such multiple enhancers often play different functional roles, while it is also known that they sometimes show overlapping functions (Osterwalder et al., 2018). […] Our observation that there are overlapping functions of CNS17, CNS20, and CNS35 in the regenerating nephric tubules, albeit with differential activation mechanisms, suggests that there is a redundancy of *cis*-regulatory elements for regeneration”.

I am confused about the Arid3a expression, according to Xenbase the levels of expression in kidney are similar for Arid3aL and 3aS, and in the figures presented, neither appears to be expressed in the pronephric region, though Arid3aL has an overall higher staining in the ISH than does Arid3aS. However, this is not as well controlled as the RNA seq in Xenbase, though that is for adult kidney.

Thank you for this comment. As you have pointed out, RNA-seq data from Xenbase were used for adult tissues. We apologize for this confusion. Therefore, we removed such data from this manuscript. Instead, we again performed *in situ* hybridization and provided better images. We also performed immunostaining using anti-Arid3a antibody and found that Arid3a protein was present in the proximal tubules.

The logic of the experiments in Figure 6 would be clearer if it were made clear that the photoMO is the sense strand which binds and inhibits the antisense.The results with the photoactivable MO appear quite convincing, though I am still puzzled by how it targets the pronephros, when the evidence for arid3aL expression is not so clear.

Thank you for this comment. We apologize for this confusion. Due to the fact that we do not have any method of targeting UV irradiation to the pronephric duct, we treated whole embryos with UV. This indicates that most of the cells in the tadpoles were irradiated. However, we were able to control the timing of UV irradiation. We have thus amended the main text as follows: “Since Photo-MO is cleaved by ultraviolet (UV) treatment, we were able to control the timing of gene knockdown.”

Reviewer #3:[…] 1) What distinguishes the Regeneration Signal Responsive Enhancers described in this work from developmental regulatory elements in the lhx1 locus? i.e. are these elements the same as cis-elements active during development? Since the fish to human enhancer conservation was stronger than the fish to frog enhancer conservation, this seems to suggest the RSREs might simply be developmental enhancers reactivated by injury, raising the question of whether they warrant the name "regeneration signal response enhancers". Also, wouldn't it have been more interesting to look at the CNEs that were specific to fish and frog (which can regenerate), even if they were not as strong, and look for binding sites in those that were not present in mouse/human (which don't regenerate as well)?

Thank you for this comment. In order to address this point, we performed transgenic reporter experiments of RSREs in embryos at the early tailbud stage. This transgenic reporter analysis showed that the RSREs CNS20 and CNS35 have no enhancer activities in developing pronephros. Although some of the CNS17 transgenic reporter embryos showed GFP expression in developing pronephros, most did not. These results suggest that the enhancers that we identified on the basis of the activities in regenerating nephric tubules primarily function after nephrectomy to induce *lhx1*, although it is still difficult to determine definitively whether RSREs are specific for regeneration (Supplementary Figure 3—figure supplement 1).

As you have pointed out, it is interesting to investigate unique binding motifs that are only commonly found in fish/frog CNSs. These unique transcription binding motifs may provide the unique trait of high regenerative capacity in fish and frog. Although this possibility is very interesting, we focused here on strongly activated enhancers. As such, instead, we included the following text in the “Discussion” section: “In addition, we also found that many CNSs that are only conserved between frog and fish showed weak enhancer activities in regenerating nephric tubules (Figure 3C). […] Further studies of transcriptional mechanisms for frog/fish CNSs may provide novel insights into the unique trait of high regenerative capacity in fish and frog”.

2) As described by the authors, the rationale for choosing to focus on lhx1a as a regeneration gene conflates two different repair mechanisms in frog and fish when the authors say "Among these, the previous finding that lhx1a-expressing cells regenerate functional nephrons in zebrafish prompted us to focus on the mechanisms of Lhx1 expression in regenerating [frog] nephric tubules (Diep et al., 2011)." The zebrafish repair mechanism involves Six2-expressing adult kidney stem cells while the frog nephrectomy model appears to be a regrowth of an existing tubule, much like mammalian kidney regeneration/wound healing, and this tubule does not express Six2. It is important to be clear about what each model represents in terms of distinct regenerative mechanisms.

Thank you for this comment. As you have pointed out, the regeneration of nephric tubules in frog appears to involve the regrowth of existing tubules. This regenerative mechanism is much closer to the regeneration of nephric tubules in mammals, rather than the reconstruction of nephrons in zebrafish from stem cells. We have thus amended the “Results” section as follows: “It has been shown in a previous study involving simple mRNA injection using *Xenopus* embryos that Lhx1 with Pax8 can induce ectopic pronephron structures (Carroll and Vize, 1999). […] These previous findings prompted us to focus on the mechanisms of *lhx1* expression in regenerating nephric tubules”. We have also included the following text in the “Discussion” section: “In zebrafish, *lhx1a* and *six2*mesenchymal cells reconstruct the functional nephrons (Diep et al., 2011). […] Further live imaging studies using transgenicanimals with markers of dedifferentiating cells and regrowing cells are required in order to reveal the detailed regenerative mechanisms of nephric tubules in vertebrates”.

3) Have the authors tried other unbiased approaches to identifying proteins bound to RSREs?

We agree that unbiased approaches to identify the enhancers are crucially important. Since we have established the identification of enhancers that are activated in regenerating nephric tubules in this research, we would like to apply unbiased approaches combining ATAC-seq and transgenic reporter analysis in the future.

4) In Figure 4D, there is no explanation of why CNE35 was split into 56-334 and 314-580.

Thank you for this comment. The sequence length of CNS35 is 580 bp. Since this is too long for qPCR, CNE35 was divided into two segments. We have included the following text to explain this: “CNS35 was divided into two segments for qPCR, since it is 580 bp long, which is too long for qPCR”.

5) In Figure 5C why would CNE35 act differently from CNE17 and CNE20 in terms of altered chromatin structure?

As you have pointed out, the chromatin configuration of CNS35 differs from that of CNS17 and CNS20, and the luciferase reporter analysis also showed that the mechanism of activation of CNS35 also differs from those of CNS17 and CNS20 (Figure 4D). At present, we do not have any explicit data to explain the difference between CNS35 and other RSREs. Therefore, we have included the following text in the “Discussion” section: “CNS17, CNS20, and CNS35 are activated in regenerating nephric tubules, indicating that these enhancers have overlapping functions. […] Our observation that there are overlapping functions of CNS17, CNS20, and CNS35 in the regenerating nephric tubules, albeit with differential activation mechanisms, suggests that there is a redundancy of *cis*-regulatory elements for regeneration”.

6) From the data shown I am not convinced Arid3a is expressed in the pronephros. The magenta arrows in Figure 4B seem to be pointing at yolk with no blue *in situ* reaction product where the pronephros would be. The cross section is too low resolution to see tubule morphology in the stained cross section. Clearer data should be presented.

We performed *in situ* hybridization again and provided better images. We also performed immunostaining using anti-Arid3a antibody and found that Arid3a protein was present in the proximal tubules. We have thus amended the text as follows: “*arid3a* is known to be expressed in the ectoderm of the early neurula and in the epidermis at the late tailbud stage, whereas *spib* is expressed in the anterior ventral blood islands at the neurula stage (Callery et al., 2005; Costa et al., 2008). […] In addition, we performed immunostaining of Arid3a using *Xla.Tg(Xtr.pax8:EGFP)*, whichshowed that Arid3a protein was detected in proximal tubules and also in the glomus and/or nephrocoelom (Figure 4B, orange arrows and orange arrowheads, respectively)”.

7) In Figure 4C, the experiment is a little difficult to understand. How many hours after heat shock was lhx1 expression assayed?

Thank you for this comment. We have added the experimental design in Figure 4C.

8) Figure 6: What accounts for the loss of message in UV treated photo MO condition? Splice blocking MOs usually alter splicing; is there an altered Arid3a mRNA after UV activation of the MO? Is there a control for UV exposure alone on mRNA abundance? 30 minutes of continuous exposure to 365 nm light could have effects of its own. Also the experimental design at the top of 6A should indicate that both photo and regular MO's were injected together.

Thank you for this comment. We designed the splicing-blocking morpholino to the *arid3a.L* exon2-intron2 donor site and also designed RT-PCR primers to exon 2 and exon 4. Therefore, exon 3 of *arid3a.L* is skipped by *arid3a.L-*MO, and the level of exon 3-containing transcripts is decreased. In order to confirm whether UV exposure affects mRNA abundance, we performed RT-PCR using UV-untreated and UV-treated wild-type tadpoles. This analysis showed that UV does not affect the abundance of spliced mRNA. We added the result on this in Author response image 1.

**Author response image 1. respfig1:** A) A diagram of the experimental design. B) RT-PCR analysis was performed. Lanes 1: Marker, Lane2: UV untreated embryos, Lane3: UV treated embryos. UV treatment and RT-PCR were performed at three times independently. C) The significance of differences between the UVuntreated and UV-treated wiled type embryos was calculated by two-tailed unpaired t-test (*p* = 0.4857, not significant). The error bars indicate SEM.

We have also changed Figure 6A as suggested and included the following text: “We first designed the splicing-blocking morpholino for *arid3a.L*, since the expression *arid3a.L* in nephrons is stronger than that of *arid3a.S* (Figure 4B, Figure 4—figure supplement 1A). […] Thus, *arid3a.L-*MO is rendered inactive by binding to *arid3a.L*-Photo-MO”.